

# Climate change, re-/afforestation, and urbanisation impacts on evapotranspiration and streamflow in Europe

Adriaan J. Teuling[1], Emile De Badts[1], Femke A. Jansen[1], Richard Fuchs[2], Joost Buitink[1], Anne J. Hoek van Dijke[1,3,4], and Shannon Sterling[5]

[1]Hydrology and Quantitative Water Management Group, Wageningen University & Research, Wageningen, The Netherlands
[2]Karlsruhe Institute of Technology (KIT), Institute of Meteorology and Climate Research, Atmospheric Environmental Research (IMK-IFU), Group of Land Use Change and Climate, Garmisch-Partenkirchen, Germany
[3]Luxembourg Institute of Science and Technology, Belvaux, Luxembourg
[4]Laboratory of Geo-information Science and Remote Sensing, Wageningen University & Research, Wageningen, The Netherlands
[5]Department of Earth Sciences, Dalhousie University, Halifax, Canada

**Correspondence:** Adriaan J. Teuling (ryan.teuling@wur.nl)

**Abstract.** Since the 1950s, Europe has seen large shifts in climate and land cover. Previous assessments of past and future changes in evapotranspiration or streamflow have either focussed on land use/cover or climate contributions, or have focussed on individual catchments under specific climate conditions. Here, we aim to understand how decadal changes in climate (e.g., precipitation, temperature) and land use (e.g., de-/afforestation, urbanization) have impacted the amount and distribution of water resources availability across Europe since the 1950s. To this end, we simulate the distribution of green and blue water fluxes at high-resolution (1 km$^2$) by combining a) a steady-state Budyko model for water balance partitioning constrained by long-term (lysimeter) observations across different land-use types, b) a novel decadal high-resolution historical land use reconstruction, and c) gridded observations of key meteorological variables. The continental-scale patterns in the simulations agree well with coarser-scale observation-based estimates of evapotranspiration, and also with observed changes in streamflow from small basins across Europe. We find that strong shifts in the continental-scale patterns of evapotranspiration and streamflow have occured from 1950 to 2010. In Sweden, for example, increased precipitation dominates effects of large scale re- and afforestation leading to increases in both streamflow and evapotranspiration. In most of the Mediterrenean, decreased precipitation combines with increased forest cover and potential evapotranspiration to reduce streamflow. In spite of local and regional scale complexity, the Europe-wide net contribution of land use, precipitation and potential evapotranspiration changes to changes in ET is similar with ∼40 km$^3$/y, equivalent to the discharge of a large river. For streamflow, changes in precipitation dominate land use and potential evapotranspiration contributions with ∼90 km$^3$/y compared to ∼45 km$^3$/y. Locally, increased forest cover and urbanisation have lead to significant decreases and increases of available streamflow.



# 1 Introduction

Streamflow provides an integrated signal both in space and time of all upstream changes in the terrestrial hydrological cycle. At smaller timescales of days to weeks, streamflow reflects the weather conditions and precipitation in the recent past. At longer (multi-year) timescales, over which internal catchment storage changes become much smaller than the amount of water

passing through the catchment system, streamflow reflects the amount of water that passes through aquifers and dams (the "water yield"), which is the portion of precipitation that is not returned to the atmosphere via evapotranspiration (the so-called "green water" flux, here used to indicate the total evaporative flux). The water yield represents the average water flux that can potentially be exploited for human benefit in a sustainable way, and is nowadays often referred to as "blue water" availability or "blue water" flux. Quantifying and understanding past and future changes in blue water availability, the integrated signal of

all net changes in the water cycle upstream, is not only of key importance to water resources management and planning, it is also a major scientific challenge given the uncertainties and limitations in both observations and models. This is in particular true for Europe, where strong changes in land use (in particular urbanisation, re- and afforestation, see Fuchs et al., 2013), and climate (van der Schrier et al., 2013; Caloiero et al., 2018; Fontrodona Bach et al., 2018) have occurred since the 1960s. In the following, we will use several terms interchangeably: green water flux or evapotranspiration (ET) to describe to the total

average evaporative return flux to the atmosphere (including interception evaporation), and blue water flux, water yield, or streamflow to describe the average flux of water from a land area (although it should be noted that not every pixel considered might be directly connected to a stream).

Several studies have focussed on large-scale changes in green and/or blue water fluxes. In one of the first global studies, Milly et al. (2005) analysed climate-driven changes in blue water availability from an ensemble of climate models and found

a general drying of transitional regions and a wetting of current humid and colder regions. Over Europe, the study reported a strong latitudinal gradient in average blue water fluxes increasing in strength from the $20^{th}$ to the $21^{st}$ century, with decreasing trends in the Meditarrenean, and increasing trends in Northern Europe. Gerten et al. (2008) showed that globally, precipitation changes were the biggest drivers of changes in runoff, but also land use change had a considerable effect. Changes in Northern Hemisphere streamflow over the past decades have likely also been impacted by decadal changes in solar radiation (the so-

called global "dimming" and "brightening", see Teuling et al., 2009; Gedney et al., 2014). Other studies have focussed on the impact of anthropogenic land-cover change on green water fluxes. Sterling et al. (2012) found a 5% reduction in global ET due to land cover conversion, resulting in a 7.6% increase in global average streamflow. Other studies have highlighted strong decadal-scale variability in global average ET over the recent decades related to El Niño–Southern Oscillation (Jung et al., 2010; Miralles et al., 2014). In spite of the direct link between average green and blue water fluxes, few studies have addressed

changes in both fluxes simultaneously.

Because streamflow is impacted by many factors, which often have opposing effects, changes in streamflow should be considered at small scales at which indivual factors can be understood and quantified rather than at larger river basin scales. Although several long discharge records exist for large river basins, changes that occur at the sub-basin level are often obscured by opposing effects in other parts of the basin. In a landmark study, Stahl et al. (2010) addressed this limitation by analysing



streamflow changes in Europe from a dataset of relatively small river basins with limited human influence. They reported a diverging pattern of streamflow trends over the past decades, with negative trends in annual mean streamflow in many parts of the Mediterrenean and Central Europe, and predominantly positive trends in Western Europe and parts of Scandinavia. While the longer-term and long-range variability of streamflow in these basins and its relation to circulation indices is generally

well understood at the interannual and decadal timescales (Gudmundsson et al., 2011; Hannaford et al., 2013), significant uncertainty exists in understanding the regional-scale variability in trends since these are not well reproduced by the current generation of hydrological models (Stahl et al., 2012). Previous case studies for catchments across Europe have reported a sensitivity of long-term water balance partitioning to both climate and land use change (Parkin et al., 1996; van Roosmalen et al., 2009; van der Velde et al., 2013; Pijl et al., 2018). Thus, quantifying changes in streamflow requires accounting for

changes in climate (precipitation and potential evapotranspiration) as well as changes in land use and/or land cover (Stonestrom et al., 2009). But whereas assessing the impact of climate on average streamflow is relatively straightforward, the role of land cover requires a more careful consideration.

At the smaller scale, land use, in particular forest cover, has longsince been known to have a strong impact on average streamflow or water yield, with forested catchments having a much lower water yield compared to non-forested catchments

(Bosch and Hewlett, 1982; Zhang et al., 2001; Brown et al., 2005; Farley et al., 2005; van Dijk and Keenan, 2007; Filoso et al., 2017). Based on analysis of paired catchment observations, a large majority of studies have found that removal of forest leads to an increase in water yield. While this is likely linked to higher average evapotranspiration over forest, the reverse has been reported for dry and warm summer conditions based on eddy-covariance observations from FLUXNET (Teuling et al., 2010). Somewhat surprisingly, average evapotranspiration rates for forested FLUXNET sites are on generally slightly lower than for

non-forested sites (Williams et al., 2012), which is seemingly inconsistent with other studies (e.g., Zhang et al., 2001) where annual evapotranspiration was inferred from the water balance (the so-called "forest evapotranspiration paradox", see Teuling, 2018). A possible explanation for this discrepancy is the role of interception (van Dijk et al., 2015). Several studies (e.g., Gash et al., 1980; Zimmermann et al., 1999) have shown that interception can constitute a major term in the water balance of forested ecosystems, in particular in humid conditions (Calder, 1976; Ramírez et al., 2018). Controlled experiments on large

non-weighable lysimeters covered with forest have shown that growing forest strongly reduces the water yield (Tollenaar and Ryckborst, 1975; Harsch et al., 2009; Müller, 2009; Teuling, 2018), and that this effect is somewhat larger for coniferous than for deciduous species.

In contrast to forest cover, effects of urban area and urbanization on the long-term water balance partitioning are largely unknown. Runoff from urban areas is typically measured with focus on event runoff ratios (Berthier et al., 1999) or runoff

produced by impervious areas only (Boyd et al., 1993; Shuster et al., 2005). Evapotranspiration from urban areas is typically measured or analysed over individual elements that make up the urban landscape, such as (un)paved areas (Ramamurthy and Bou-Zeid, 2014), green roofs, or trees (Pataki et al., 2011). Few studies have measured ET at the urban landscape scale. In a study comparing measurements made over the Dutch cities of Rotterdam and Arnhem, Jacobs et al. (2015) found ET rates to be generally low, and to quickly drop in the days following rainfall reflecting a strongly water-limited system. This suggests



that urban areas, because of their limited capacity to store water, might have much lower green water fluxes and as a result much higher blue water fluxes than other land use types.

In order to isolate the hydrological impact of climate change from that of changes in land use, models with varying levels of complexity are used. Typically, hydrological or land surface models run at hourly or daily resolution are used (Bosmans et al., 2017; Breuer et al., 2009; Viney et al., 2009; Dwarakish and Ganasri, 2015; Pijl et al., 2018). Such models often contain a high number of poorly-constrained parameters and parameterizations, leading to large uncertainty in trend estimates (Arnell, 2011), or even disagreement in the direction of simulated trends (Melsen et al., 2018). When the research focus is on robust simulation of long-term rather than short-term changes, low-dimensional models with well-constrained parameters often perform well (Choudhury, 1999; Zhang et al., 2008). The Budyko model (Budyko, 1974) is an example of such a model which allows for evaluating combined land use and climate impacts on water availability. In spite of its extreme simplicity (parameterizations typically have only one parameter reflecting land surface characteristics), it has been applied successfully in numerous studies focussing on different controls on long-term water balance partitioning (Zhang et al., 2004; Roderick and Farquhar, 2011; Xu et al., 2013; Greve et al., 2014; Xu et al., 2014; Creed et al., 2014; Jiang et al., 2015; Zhang et al., 2016; Wei et al., 2018). Although it is generally applied at coarse global grid resolution or to large river basins, other studies (e.g. Zhang et al., 2004; Redhead et al., 2016) have found the model to also work well for smaller basins or grid cells ($< 10$ km$^2$). This opens up the possibility for robust and parsimonious modeling of hydrological impacts at high spatial resolution.

The strong impact of land use on water balance partitioning at smaller scales, combined with the large-scale land use changes that have occurred over Europe over the past decades, leads to the question how they have impacted changing patterns of green and blue water fluxes. Previous assessments of past and future changes in streamflow have either focussed on land use/cover (Sterling et al., 2012) or climate contributions (Wilby, 2006; Gardner, 2009; Hannaford et al., 2013), or have focussed on smaller catchments under particular climate conditions (van Roosmalen et al., 2009; Pijl et al., 2018). Therefore, we aim to understand how recent decadal changes in climate (e.g., precipitation, temperature) and land use (de-/afforestation, urbanization) have impacted the amount and distribution of water resources availability across Europe since the 1950s. We address the hypothesis that land cover changes play a much more important role at the European scale than previously reported, even in basins which are assumed to have a limited human influence on the water cycle. To this end, we simulate the distribution of green and blue water fluxes at high-resolution (1 km$^2$) by combining a) a steady-state Budyko model for water balance partitioning constrained by long-term observations across different land-use types, b) a novel decadal high-resolution historical land use reconstruction, and c) gridded observations of key meteorological variables. Simulations will be evaluated against state-of-the-art observation-based assessments of evapotranspiration and observed changes in streamflow.

## 2   Methods and Data

Central to our approach is the formulation of the Budyko model as used by Zhang et al. (2004). As with any Budyko approach, it follows the central assumption that the fraction of precipitation that returns to the atmosphere as evapotranspiration ET depends on the ratio between the average potential evapotranspiration PET and average precipitation $P$, rather on their absolute



values. Good fit with observations at several spatial scales show that this assumption is justified. Here, we will calculate PET by the Thornthwaite method (see, e.g., van der Schrier et al., 2011). This method only requires temperature as input, and as a result it is not sensitive to changes in other variables affecting evaporation, such as vapor pressure deficit, wind speed, or net radiation. While the benefit of this approach is that the analysis can be carried out beyond the record for routine incoming shortwave

and/or net radiation observations, it has as main drawback that temperature is not always a reliable proxy for radiation in particular under global warming. The potential implications of this assumption are discussed in Section 4.

In the work by Zhang et al. (2004), the following equation was proposed for the dependency of $\text{ET}/P$ on $\text{PET}/P$:

$$\frac{\text{ET}}{P} = 1 + \frac{\text{PET}}{P} - \left[ 1 + \left( \frac{\text{PET}}{P} \right)^w \right]^{1/w} \tag{1}$$

in which $w$ is a model parameter which is typically linked to catchment and/or vegetation properties (Li et al., 2013). Zhang

et al. (2004) found $w = 2.63$ to best fit observations for Australian catchments, with slightly lower values for grassed ($w = 2.55$) and higher for forested catchments ($w = 2.84$). While these different values confirm that $w$ depends on land surface characteristics, the magnitude of this effect at the scale of individual land use elements is probably larger. Based on analysis of remotely sensed Normalized Difference Vegetation Index (NDVI) and gridded global fields of ET, PET, and $P$ at the $0.5°$ resolution, Greve et al. (2014) reported values of 3.05 for grid cells with an NDVI of around 0.8, whereas grid cells with an

NDVI of around 0.2 where found to follow $w = 1.63$. In a similar study but using observed streamflow rather than estimated $E$, Li et al. (2013) found $w$ to depend on the basin-average fractional vegetation cover $M$ according to $w = 2.36 \times M + 1.16$. These studies show that $w$ can show considerable variation even at relatively course scales.

In order to get the most realistic values for $w$ for application at smaller scales ($\sim 1 \text{ km}^2$) at which land use is often fairly homogeneous and the effects on water balance partitioning are most pronounced, we constrain $w$ by the best available ob-

servations for different land use types and made under European climate conditions. It should be noted that widely available FLUXNET observations are not used in this study, because they might show the opposite land use ET signal from water balance-based studies (the so-called forest evapotranspiration paradox, see Teuling, 2018) which are assumed to be more reliable for long-term water balance analysis. The observations used in this study primarily come from the long-term lysimeter stations, such as the ones at Rietholzbach (Seneviratne et al., 2012), St. Arnold (Harsch et al., 2009), Brandis (Haferkorn and

Knappe, 2002), Eberswalde-Britz (Müller, 2009), Castricum (Tollenaar and Ryckborst, 1975), and Rheindahlen (Xu and Chen, 2005), several of which were also analysed in a previous study by Teuling (2018). This data is complemented by observations from a natural lysimeter at Plynlimon (Calder, 1976) under more humid climate conditions and flux observations made over the cities of Arnhem and Rotterdam (Jacobs et al., 2015). Long-term data is preferred to minimise impacts of interannual storage variations (Istanbulluoglu et al., 2012). By relying on lysimeter observations to constrain our Budyko parameters, we implicitly

assume lysimeters (area 1–625 m$^2$) to behave similar to landscape elements of $10^6$ m$^2$ (our grid cell size). The data is shown in Fig. 1 and listed in Table 1. It should be noted that the stations are not distributed evenly across Europe, but are mainly constrained to Central-Western Europe (Fig. A1).

Due to the smaller scale than applied in previous Budyko analyses, we initially find many points, in particular observations from forested lysimeters, to be located above the energy-limit (grey dashed line in Fig. 1). This indicates that the long-term




average yearly evapotranspiration (ET) exceeds the average potential evapotranspiration (PET). This is possible, for instance, due to evaporation of interception water by energy not captured in the formulation of PET (van Dijk et al., 2015). Therefore, we correct for the underestimation by introducing a so-called adjusted potential evapotranspiration (aPET) which is assumed to be proportional to the potential evapotranspiration and accounts for all processes affecting yearly ET for tall vegetation (including

evaporation of intercepted water through advection):

$$\text{aPET} = c \times \text{PET}, \tag{2}$$

resulting in the following expression for the Budyko-curve:

$$\frac{\text{ET}}{P} = 1 + \frac{\text{aPET}}{P} - \left[ 1 + \left( \frac{\text{aPET}}{P} \right)^{w^*} \right]^{1/w^*} \tag{3}$$

in which $w^*$ is the value for $w$ when aPET rather than PET is used. The values for $w^*$ that match the (lysimeter) observations

are shown in Fig. 1. It was found that $c = 1.8$ was required to ensure all observations would be located on the right-side of the energy-limit ($\text{ET}/P = \text{PET}/P$). Subsequently, in all analysis we replaced PET with aPET, including Eq. 1 but also in the atmospheric forcing fields. It should be noted that while this procedure results in lower values for $w^*$ that cannot be directly compared to values for $w$ reported in previous studies, most of the simulated ET values are identical to the ones that would be simulated with the original model. We find the highest $w^*$ for full-grown forest, indicating that any change towards this state

due to re- or afforestation will increase ET given the same climate ($P$ and PET). Conversely, urban areas have low $w^*$, indicating that urbanisation will generally decrease ET. Finally, the long-term average streamflow or blue water flux at the pixel level is calculated from the catchment water balance:

$$Q \approx P - \text{ET} \tag{4}$$

under the assumption that storage changes (such as snow, soil moisture, groundwater) and lateral transport between can be

neglected at the decadal (10-year) timescale. This timescale is chosen to align with the temporal resolution of the land use dataset, and to minimize possible impacts of storage changes.

As input to our model as described above (Eqs. 2–4), we use gridded datasets of land cover and meteorological observations. All calculations where performed at a $1 \times 1$ km spatial resolution, which were later rescaled to a coarser resolution for visualization purpose. Historic land-change information is based on the HIstoric Land Dynamics Assessment (HILDA, v2.0) model

reconstruction of historic land cover/use change (Fuchs et al., 2013, 2015a, b). This data-driven reconstruction approach used multiple harmonized and consistent data streams such as remote sensing, national inventories, aerial photographs, statistics, old encyclopedias and historic maps to reconstruct historic land cover at a $1 \times 1$ km spatial resolution for the period 1900 to 2010 in decadal time steps. The reconstruction provides information for six different land cover/use categories: forest, grassland (incl. pastures, natural grasslands and shrublands), cropland, settlements/urban, water bodies and other (i.e. bare rock, glaciers

etc.). Here we only use the forest, grass-/cropland, and settlement classes. The reconstruction considers gross land changes, the sum of all area gains and losses that occur within an area and time period, unlike other reconstructions that focus on net



changes only, calculated by area gain minus the area losses. Details on the net versus gross changes can be found in Fuchs et al. (2015a). The gross changes are used to derive forest stand age. We distinguish between young stands (age < 10 years, assumed to behave similar to crop-/grasslands), intermediate (age 10–20 years) and older stands (age > 20 years), see also Fig. 1. Previous research has shown that not accounting for gross land use changes in reconstruction led to serious underestimations in

the amount of total land use changes that have occurred (Fuchs et al., 2015a). The E-OBS v17 gridded datasets (Haylock et al., 2008) of observed precipitation and temperature were used to force the model (Eq. 3). Temperature was used to calculate PET according to the Thorntwaithe method. Based on the joint availability of both datasets, we selected two 10-year periods which were considered for analysis: 1955–1965 and 2005–2015. In the following, we will refer to these periods as 1960 and 2010 for simplicity. Changes over the intermediate 10-year periods were analysed, but since the trends were generally found to be

monotonic the results are not shown here (except for validation porpuses in Fig. 5).

Model simulations are validated against observed yearly average streamflow changes in near-natural catchments and observation-based average evapotranspiration. The relative streamflow changes for the period 1962–2004 (normalized by the standard deviation of yearly streamflow) were taken directly from Stahl et al. (2010, their Figure 2). Average evapotranspiration was derived from GLEAM v3.2a (Martens et al., 2017). The contribution of $P$, PET and land use (through $w^*$) was assessed by perform-

ing separate simulations in which only one of the three factors was varied while the others were kept constant at their 1960s reference.

## 3   Results

Recent changes in climate have lead to substantial changes in the magnitude and distribution of precipitation and potential evapotranspiration, the two main climate drivers in the Budyko model (Eq. 1) that determine how average precipitation is

partitioned between evapotranspiration and streamflow. Average precipitation during the reference period shows a general decrease towards the East (Fig. 2a). Superimposed on this large-scale pattern are local areas with higher precipitation along the coastal areas in the West and/or in mountainous regions. Changes in average precipitation over the study period show a strong North-South gradient (Fig. 2b): Most of the Mediterrenean, in particular the Iberina Peninsula, shows a decline in precipitation, whereas Northern Europe, in particular the British Isles, the Scandinavian Peninsula and Finland, have seen strong increases

in average precipitation regionally exceeding 20%. In contrast to precipitation, potential evapotranspiration shows a strong latitudinal gradient (Fig. 2c) with lower values (PET around 400 mm/y) in Scandinavia and higher (around 700 mm/y) in the Mediterrenean. Changes in potential evapotranspiration (Fig. 2d) are predominantly positive (decreasing values in Romenia likely reflect a data-quality issue) and highest in Central Europe reflecting the higher increase in average temperatures. In general, these strong changes in climate forcing ($P$ and PET) are likely to be reflected in continental-scale patterns of changes

in water availability.

In addition to climate, also land use and land cover in Europe have seen large scale shifts over the past 60 years, albeit on a more local scale. Figure 3 shows the mean forest and urban fraction for the reference period, as well as the fractional change over the period 1960–2010. While forest cover is widespread over most of Europe (Fig. 3a), most extensive forest regions





can be found in central-western Europe, Sweden and Finland. Forest cover has increased considerably over most of Europe (Fig. 3b) following abandonment of less-productive agricultural areas and intensification of forestry and forest management, with Sweden (Ericsson et al., 2000) and the Mediterrenean region showing the strongest changes. It should be noted that areas where forest cover has declined are virtually absent. This is also true for change in urban area. The average urban fraction is

5 highest in central-western Europe (Fig. 3c), and this is also the region that has seen the strongest increased (Fig. 3d). Changes in urban area are generally more localized in nature than changes in forest cover.

Patterns of mean and changes in evapotranspiration and water yield were calculated by forcing the Budyko model with subsequent 10-year averages of climate forcing and land use at a $1 \times 1$ km resolution. Figure 4 shows the resulting continental-scale patterns. The mean evapotranspiration in the reference period (Fig. 4a) is highest in central Europe, locally exceeding

800 mm/y, in regions with pronounced topography and/or forest cover. The Nordic countries and the Iberian Peninsula generally have lower values (<400 mm) due to more pronounced energy and water limitation, respectively. Changes in evapotranspiration show a strong latitudinal gradient (Fig. 4b). Changes exceeding +15% are found in large parts of Scotland, Sweden, Finland, and Estonia, whereas most of Central-Western Europe shows a smaller increases in the order of 10%. Decreases of similar magnitude occur in parts of the Iberian Peninsula and Italy. Average streamflow (Fig. 4c) is highest in Central-Western Europe

(locally exceeding 600 mm/y), in particular in mountainous areas that receive larger amounts of precipitation. Streamflow of less than 150 mm/y is found in the large parts of Sweden, Finland, Spain, Rumenia and Bulgaria. Changes in water yield (Fig. 4d) show a roughly similar pattern to changes in evapotranspiration, however the changes are much stronger in magnitude. Decreases in the Mediterrenean locally exceed −45%, where increases in Sweden and Finland exceed +45%. Both the changes in evapotranspiration and streamflow show considerable regional variability superimposed on the large-scale patterns.

In order to assess the quality of the simulated evapotranspiration and streamflow and the changes therein, we evaluate our simulations against observation-based estimates of average evapotranspiration (Martens et al., 2017) over the more recent period 1980–2017 (it should be noted that currently no gridded evapotranspiration estimates are available that cover our complete study period) as well as observed changes in streamflow reported by Stahl et al. (2010) that cover most of our study period. The pattern of simulated ET (Fig. 5a) closely resembles the pattern as produced by GLEAM version 3.2a (Martens et al., 2017, data

shown in Fig. 5b). The Budyko model produces slightly lower values in Eastern Europe and the Iberian Peninsula, but slightly higher values in Sweden and Finland. At the regional scale, our simulations show more variability due to the higher resolution of the forcing and land use datasets. In addition to matching the pattern of average ET, our approach is also able to reproduce the overall pattern of observed changes in streamflow (Fig. 5c,d). In spite of the difference in units and the fact that individual basins might have shorter record lenghts, the correlation in trends between the basins is 0.34. The simulations agree with the

observed declines in average streamflow in much of Southern and Central Europe, and increases in the more mountainous, coastal and/or Northern regions. Overall, the validation shows that our simplified approach is able to capture continental-scale patterns in mean and changes of blue and green water fluxes.

Changes in fluxes are driven by local changes in climate and land use. Figures 6 and 7 show how the contribution of the main drivers (precipitation, PET, and land use) to changes in evapotranspiration and streamflow, respectively, vary across Europe.

This is done by plotting each contribution (as determined from simulations where the other drivers where kept constant) as a





seperate RGB component, whereby each contribution is rescaled inversely from the 2nd to the 98th percentile. The resulting colormap thus has a 3D color legend. From the distribution of colors, covering most of the possible colors, it can be readily seen that contribution of individual drivers shows a strong variability. Magenta indicates that land use-induced changes in evapotranspiration and streamflow are widespread but generally local in character. Yellow colors occur widely in a latitudinal

band between 45°N and 54°N, indicating that changes in PET have a strongest influence on water balance partitioning in transitional regions, but less so in water-limited and humid Northern regions. Finally, the relative impact of precipitation is strongest above 54°N and below 45°N. It should be noted that these continental-scale patterns differ from their changes, which are much more uniform (e.g. PET changes in Fig. 2d are fairly homogeneous).

The subpanels in Figures 6 and 7 zoom in on several regions, further illustrating the strong regional divergence in changes

in water flux partitioning. In the Southern Highlands of Scotland (Figs. 6a/7a), a strong increase in precipitation has lead to a strong net increase in streamflow of +339 mm/y, only slightly counteracted by opposing PET and land use (afforestation) effects. Urbanisation in the Paris metropolitan area (Figs. 6b/7b) has act to reduced ET, but combines with increased $P$ to a significant increase in streamflow (+44 mm/y). In the Landes forest region (Figs. 6c/7c), individual effects are small but combine to a strong (−79 mm/y) reduction in water yield. ET changes in the Seville region (Figs. 6d/7d) are relatively small

due to opposing contributions of precipitation decline and afforestation, but these effects combine into a strong reduction on streamflow (−100 mm/y). In Sweden, ET changes (Figs. 6e/6f) are stronger in the middle of the country where widespread afforestation and precipitation increase combine. As a result, increases in streamflow are stronger in the South where land use contributions do not reduce the effect of precipitation increase (Figs. 7e/7f). In Central Austria (Figs. 6g/7g), PET increases dominate the net ET change (+31 mm/y), but combine with precipitation reduction into a strong reduction of water yield

(−94 mm/y). In the Bulgarian Smolyan Province (Figs. 6h/7h), contributions combine into a strong ET increase (+69 mm/y) but largely cancel out in the net effect on water yield (−8 mm/y). The examples highlight the fact that locally, individual changes are often amplified or counteracted by other changes, but because of the water balance constraint this is only true for impacts on either evapotranspiration or streamflow.

When the results are averaged over the continental scale, land use plays a more important role than suggested by Figure 6.

Table 2 lists the Europe-wide changes in green or blue water fluxes as induced by the three main drivers. While changes in ET induced by prepitation are largest when positive and negative contributions are considered seperately, the net effect is smaller since decreases in $P$ in southern Europe are largely balanced by increases in the northern parts. As a result, net effects of land use and PET on ET are comparable to those of precipitation (around 40 km$^3$ y$^{-1}$ each), with land use having the largest contribution. These contributions correspond to nearly 1300 m$^3$ s$^{-1}$, the equivalent of the discharge of a large river. The effects

on streamflow differ slightly, with $P$ dominating both the positive and net contributions. When zooming in on the near-natural catchments used by Stahl et al. (2010), a different picture is obtained. The contribution of $P$ is less strong, likely because most of the catchments are located in Central-Western Europe where precipitation changes have been modest (Fig. 2b) compared to, for instance, Sweden. The net change in ET is mainly driven by land use and PET. For streamflow changes, $P$ is the largest net contributor at nearly 4 km$^3$, but land use contributes significantly with nearly −2 km$^3$. For individual large river basins, such

as the Rhine basin shown here, the impacts can differ significantly. Rather than precipitation, land use and PET are found to



be the main drivers of changes in blue water fluxes over the past decades. The strong sensitivity of streamflow to past land use changes seemingly contradicts the small land use effects under future land use scenarios for this catchment found in previous studies (e.g. Hurkmans et al., 2009).

## 4 Discussion

Our results are in line with many more local or regional-scale studies. In some regions, studies have found little to no trends due to dominance of natural variability on change indicators (Hannaford, 2015). For the 6.5 km$^2$ Hupsel Brook catchment in the east of the Netherlands, Brauer et al. (2018) reported no significant trend in annual runoff since the mid 1970s. In one of the few studies on long-term in situ observations of ET, Seneviratne et al. (2012) reported no significant trends of annual ET at the Rietholzbach lysimeter in north-eastern Switzerland. These findings are consistent with the results on changes in ET and

$Q$ presented in Figs. 4b and 4d. Other regions have seen negative trends. The decline in water yield in the Ebro river has been attributed to land abandonment (López-Moreno et al., 2011), whereas precipitation decline has been identified as an additional factor in most of the Iberian peninsula (Lorenzo-Lacruz et al., 2012). In Austria, increased $P$ and PET has been identified as factors driving ET increase (Duethmann and Blöschl, 2018). Also these findings are consistent with our results. This shows that even using gridded observations contain consistent information for local-scale change analysis.

The modelling approach followed here is simplified both in terms of number of model parameters, land use classes, and the parameterization of climate. While the single model parameter $w^*$ correlates with phyical land surface properties, it does not have a direct physical meaning (although expressions can be derived linking Budyko parameters to vegetation and climate characteristics, see Gerrits et al., 2009). Therefore $w^*$ might also change with mean climate conditions and/or vegetation phenology (Donohue et al., 2007), an effect that cannot be investigated due to a lack of observations in Southern and Northern

Europe. It has also been argued that the success of Budyko approaches can be partly explained by the possible adaptation of vegetation to difference in climate seasonality and soil type (Gentine et al., 2012), which would be a strong argument in favor of using such simplified models. We also use a limited number of land use classes. This number is constrained both by the limited availability of accurate estimates of long-term water balance partitioning for different land use types, as well as by the limited number of land use classes in the HILDA land use reconstruction. Nonetheless, we believe our simulations to

capture the first-order land use and climate-induced impacts. The lysimeter observations include land use with some of the highest and lowest reported ET rates, making it unlikely that we underestimate the land use-induced variability in ET. And whereas there can be considerable variability in average ET within land use classes, for instance due to vegetation and/or soil type (Haferkorn and Knappe, 2002), this variability is typically small compared to the possible range of ET over all land use classes. Our modeling approach did not explicitly consider effects other than atmospheric temperature as climate drivers of

ET. For instance, the impact of rising $CO_2$ levels on transpiration (Piao et al., 2007) were not considered. Also the impact of agricultural intensification (Liu et al., 2015) and irrigation on ET were not considered. Both can be expected to lead to higher ET and lower Q. While such processes can have strong impacts locally and regionally, other studies have shown small effect under European conditions (e.g., van Roosmalen et al., 2009). It should be mentioned that other, more rigurous, methods have





been applied at smaller scales based on multiple working hypotheses (Harrigan et al., 2014), that allow for identification of additional factors driving hydrologic change.

The model forcing is based on interpolated observations from weather stations. The location of these stations generally follows WMO recommendations, and as a result there is a lack of meteorological observations in or above forests (Frenne and
5 Verheyen, 2016), or in urban areas. Large forest or urban areas, however, are known to impact their own weather for instance due to enhanced temperature (the well-known urban-heat island effect ), cloud formation (as has been observed over the larger French forest regions of Landes and Sologne, see Teuling et al., 2017), or rainfall (as has been shown by modeling experiments for the Dutch Veluwe forest region, see ter Maat et al., 2013). Such local land cover impacts on climate are unlikely to be represented correctly in the forcing dataset used in this study which is based on interpolation of weather station data. Also the
10 quality of the data underlying the E-OBS and HILDA datasets used in this study might differ between countries. As a result, the datasets might induce "jumps" near to borders as can be seen in some of the maps. These inconsistencies will likely be fixed in future releases of the datasets, and do not impact the overall conclusions of this study.

The model forcing of potential evapotranspiration is determined by a simple temperature-based parameterization, namely the Thornthwaith method. The benefit of this approach is that simulations can be done in a consistent and robust manner for
a longer historical period. Routine observations of global radiation, needed to force more complex parameterizations such as the Penman-Monteith equation, are only available for the most recent decades from either stations or satellite. A major disadvantage of the temperature-based methods is that, while they correctly follow the intra-annual variations in energy, they might be too sensitive to interannual and decadal variations in temperature that are independent of radiation trends (Sheffield et al., 2012). In spite of the possible overestimation of the temperature effect on PET trends, we believe the impact on our main
results to be minimal in particular in (semi-arid) regions with seasonal water limitation due to the reduced sensitivity of ET to PET (van der Schrier et al., 2011). This is partly since the temperature trends in Europe have been impacted by decadal trends in radiation (i.e., global dimming and brightening, see Wild, 2016), and also because even with the possible overestimation the trends in ET and $Q$ are still dominated by changes in land use and precipitation. It has also been reported that Thornthwaite does not always gives the strongest increase in PET values in a warming climate when compared to other more physically-based
methods such as Penman-Monteith (Prudhomme and Williamson, 2013).

Changes in climate and land use generally affect both the average green and blue water flux. But whereas changes in green water fluxes are needed to explain changes in blue water, the socio-economic impact relates more directly to blue water fluxes since they reflect average fresh water availability. This is of particular relevance in the Mediterranean region, where a decline in blue water flux or streamflow reflects a decrease in water available for irrigation and agricultural production downstream.
Our results indicate that land use changes in the more mountainous areas in the Mediterrenean have contributed significantly to reductions in streamflow. Conversely, increasing blue water fluxes in Northern Europe might be beneficial to other sectors as such the hydropower industry. The finding that land use change effects are of similar magnitude as climate change effects on water availability also has important implications beyond the yearly average values. Extremes will likely also be impacted by land use, yet current drought projections for Europe (Forzieri et al., 2014; Samaniego et al., 2018) or assessments of changes
in floods (e.g. Hall et al., 2014) do not take into account past and/or future land cover changes. Not accounting for land use





change will likely lead to regional over- or underestimation of changes in water availability. Therefor, land use change impacts on green and blue water fluxes need to be considered in conjunction with climate change impacts.

## 5    Conclusions

In this study, we investigated the role of changes in land use and climate in Europe from 1960 to 2010 on average evapo-
transpiration and streamflow. In our modeling approach, we combined a state-of-the-art land use reconstruction with gridded observational datasets of climate forcing and a Budyko-model constrained with ET observations from several long-term lysimeter stations. Based on the model results, it was shown that land use changes have had net impacts on evapotranspiration that are generally comparable in size to those caused by changes in precipitation and potential evapotranspiration. Evapotranspiration increased in response to land use (mainly large-scale re- and afforestation) and climate change in most of Europe, with the
Iberian pensinsula and other small part of the Mediterrenean being exceptions with negative trends. Streamflow changes were dominated by a strong positive contribution of precipitation increases in Northern Europe. Land use and potential evapotranspiration had smaller effects of opposite sign, resulting in small net streamflow changes over Europe. The analysis revealed considerable complexity at smaller scales, with most of the possible combinations between positive and negative contributions of precipitation, land use, and potential evapotranspiration occuring at some locations. This was true for effects on evapotran-
spiration and discharge. Most pressing, we find that in much of the Mediterrenean, land use and climate change combine to further reduce blue water fluxes.

## 6    Acknowledgements

We acknowledge the E-OBS dataset from the EU-FP6 project ENSEMBLES (http://ensembles-eu.metoffice.com) and the data providers in the ECA&D project (http://www.ecad.eu). We thank Diego Miralles for providing the GLEAM data for validation.

*Data availability.*    The HILDA and E-OBS datasets are available at https://www.wur.nl/en/Research-Results/Chair-groups/Environmental-Sciences/ Laboratory-of-Geo-information-Science-and-Remote-Sensing/Models/Hilda.htm and https://www.ecad.eu//download/ensembles/download. php. All hydroclimatic observations used to constrain the Budyko model are listed in Table 1.

## Appendix A:  Appendix A

*Author contributions.*    AJT designed the study and wrote the manuscript. EDB carried out the study under supervision of AJT and FJ. RF
provided the HILDA data. All authors contributed to the writing and the interpretation of the results.



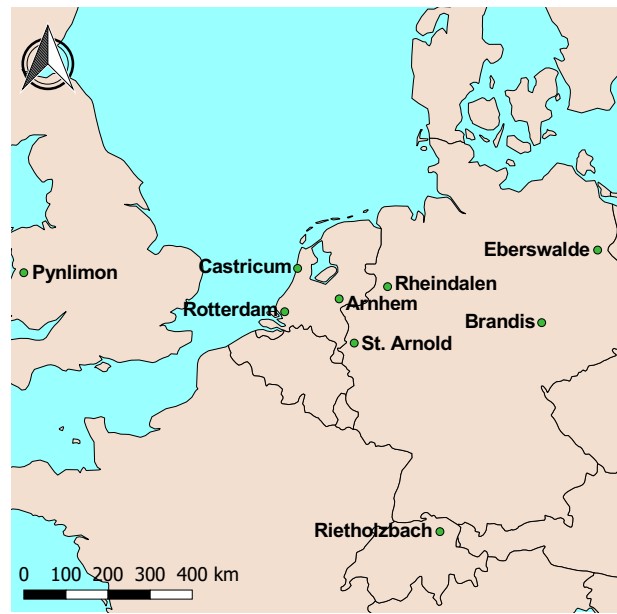

**Figure A1.** Location of the stations and sites listed in Table 1.

*Competing interests.* The authors declare no competing interests.

*Acknowledgements.* We acknowledge the E-OBS dataset from the EU-FP6 project ENSEMBLES (http://ensembles-eu.metoffice.com) and the data providers in the ECA&D project (http://www.ecad.eu).



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





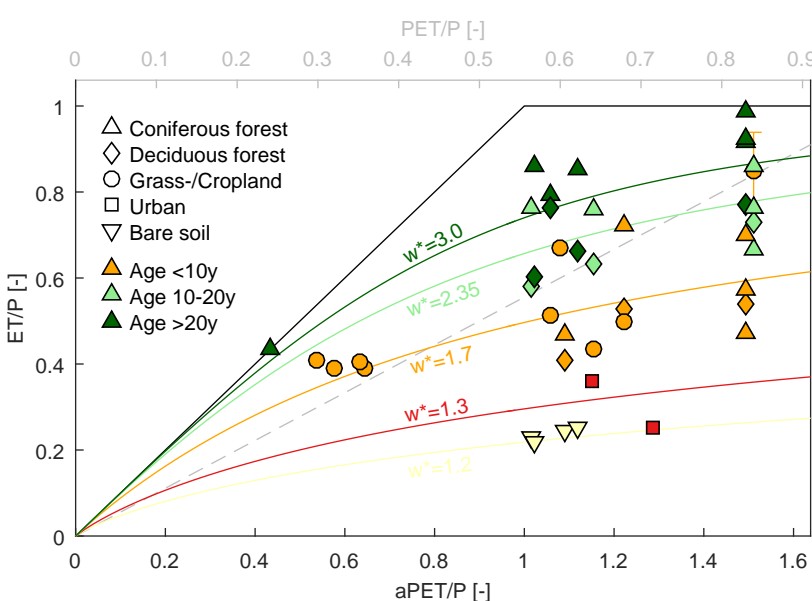

**Figure 1.** Climate and land use controls on water balance partitioning from long-term flux observations. See Table 1 for origin of data points. The errorbar indicates the total spread over multiple lysimeters at the Brandis site with different soil types (Haferkorn and Knappe, 2002). Curves are based on Eq. 3.







**Figure 2.** Climate characteristics over the period 1960–2010. Left panels show the mean precipitation and potential evapotranspiration (**a** and **c**, respectively), while the right panels indicate the change over the period 1960–2010 for precipitation (**b**) and potential evapotranspiration (**d**).



**Figure 3.** Land cover characteristics over the period 1960–2010. Left panels show the mean forest cover and urban fraction in 1960 (**a** and **c**, respectively), while the right panels indicate the change between the periods 1960 and 2010 for forest cover (**b**) and urban area (**d**).





**Figure 4.** Simulated water balance partitioning over the period 1960–2010. Left panels show the mean evapotranspiration or green water flux and streamflow or blue water flux in 1960 (**a** and **c**, respectively), while the right panels indicate the change between the periods 1960 and 2010 for evapotranspiration (**b**) and streamflow (**d**). The colors in the right panels have been chosen such that an increase in blue/green water flux also is shown in blue/green, although it should be noted that a decrease in one flux does not directly translate into an increase in the other.




**Figure 5.** Validation of simulated hydrological fluxes across Europe. **a** Observation-based ET average over the period 1980–2017 from GLEAM version 3.1 (Martens et al., 2017). **b** Simulated ET average over the 10-year periods 1990, 2000, and 2010. **c** Observed changes in streamflow over the period 1962–2004 taken from Stahl et al. (2010, their Figure 2). **d** Simulated changes in streamflow between the periods 1960 and 2000. Note the difference in units between observations (**c**) and simulations (**d**) because the approach followed in this study does not allow for normalisation by interannual streamflow variability. It should also be noted that ET validation is done for the mean flux, whereas streamflow is validated on the rate of change rather than the mean.





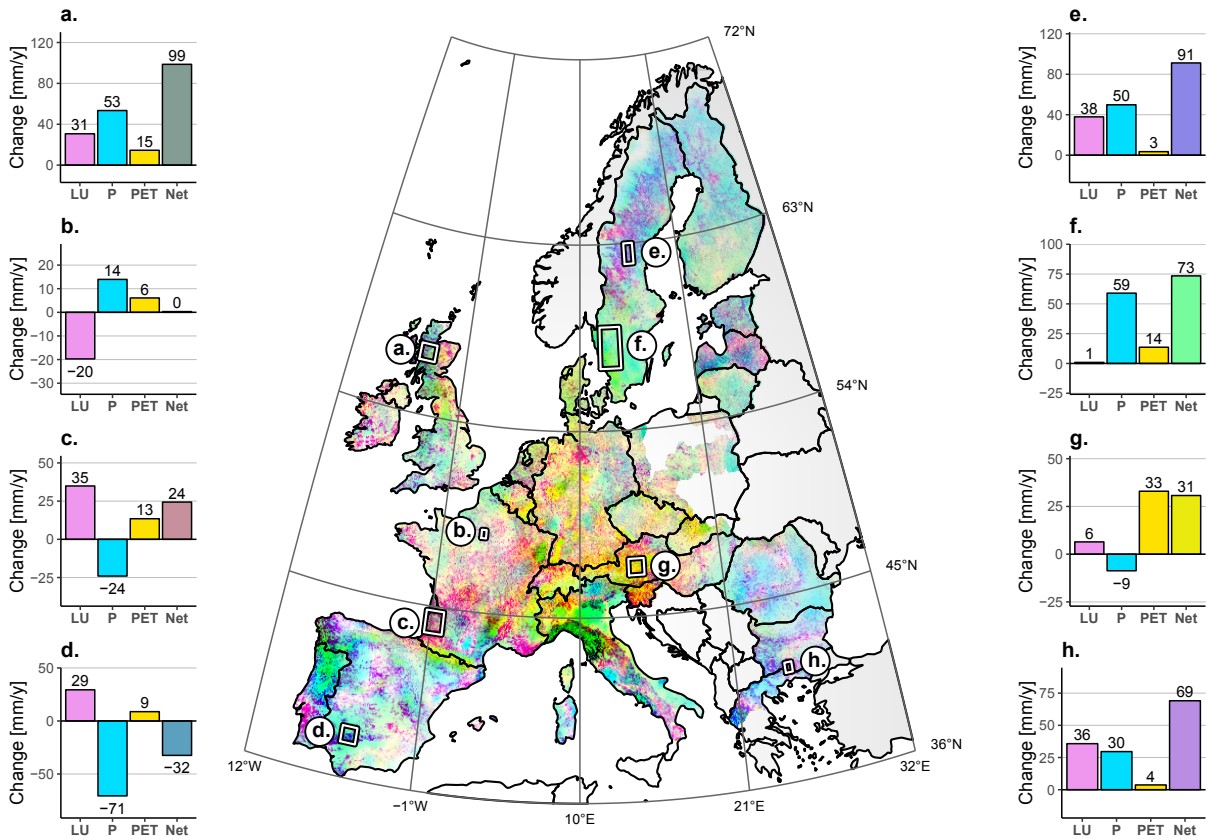

**Figure 6.** Distribution of absolute contribution of climate (P and PET) and land use (LU) changes on changes in evapotranspiration over the period 1960–2010. Colors reflect the relative importance of land use (LU, in magenta or RGB 0,255,255), precipitation (P, in cyan or 255,0,255), and PET (yellow, 255,255,0). Each contribution is inversely scaled between the 2nd and 98th percentiles to reflect its relative importance over Europe. As a result, white (255,255,255) indicates locations where all contributions are below their 2nd percentile, and black (0,0,0) indicates locations where all contributions are above their 98th percentile. Side panels show the absolute contribution of LU, P, and PET and the net change for selected regions. **a** Southern Highlands (Scotland), **b** Paris metropolitan area (France), **c** Landes forest region (France), **d** Seville region (Spain), **e** Central Sweden, **f** Southern Sweden, **g** Styria region (Austria), **h** Smolyan Province (Bulgaria). Domain-averages are listed in Table 2.





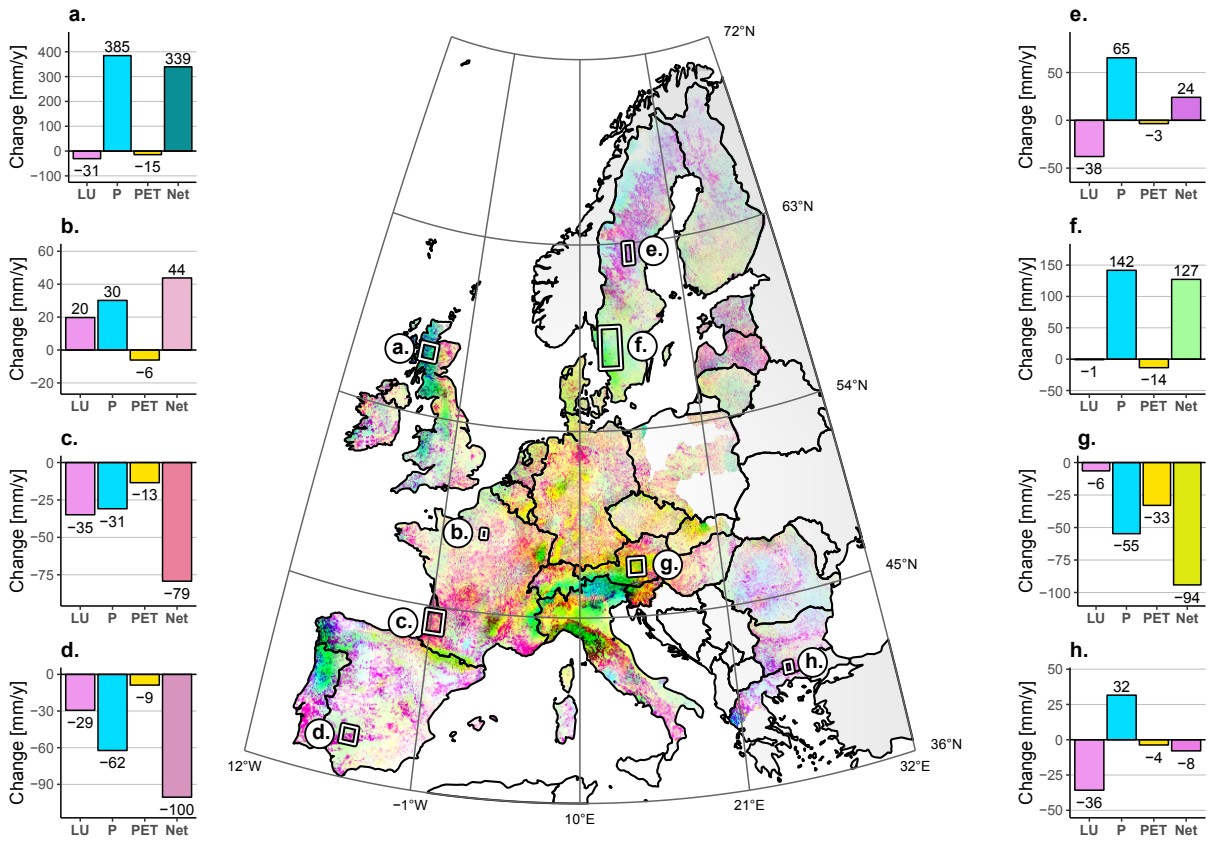

**Figure 7.** Distribution of absolute contribution of climate (P and PET) and land use (LU) changes on changes in streamflow over the period 1960–2010. See caption Fig. 5 for explanation of colors. Side panels show the absolute contribution of LU, P, and PET and the net change for selected regions. **a** Southern Highlands (Scotland), **b** Paris metropolitan area (France), **c** Landes forest region (France), **d** Seville region (Spain), **e** Central Sweden, **f** Southern Sweden, **g** Styria region (Austria), **h** Smolyan Province (Bulgaria). Domain-averages are listed in Table 2.





**Table 1.** Data used in the Budyko-analysis. Units of fluxes are in mm/y.

| Site | Lat. | Lon. | Land use | Period | $P$ | PET[*] | ET | Reference/Source |
|------|------|------|----------|--------|-----|--------|-----|------------------|
| Arnhem | 51.98 | 5.91 | Urban | 2012–2013 | 781 | 500 | 281 | Jacobs et al. (2015) |
| Brandis[**] | 51.53 | 12.10 | Cropland | 1981–1994 | 654 | 549 | 556 | Haferkorn and Knappe (2002) |
| Eberswalde | 52.89 | 13.81 | Forest (deciduous) | 1978–1984 | 633 | 525 | 341 | Müller (2009) |
| Eberswalde | 52.89 | 13.81 | Forest (deciduous) | 1985–1989 | 625 | 525 | 455 | Müller (2009) |
| Eberswalde | 52.89 | 13.81 | Forest (deciduous) | 1990–1998 | 633 | 525 | 489 | Müller (2009) |
| Eberswalde | 52.89 | 13.81 | Forest (coniferous) | 1978–1984 | 633 | 525 | 299 | Müller (2009) |
| Eberswalde | 52.89 | 13.81 | Forest (coniferous) | 1985–1989 | 625 | 525 | 417 | Müller (2009) |
| Eberswalde | 52.89 | 13.81 | Forest (coniferous) | 1990–1998 | 633 | 525 | 580 | Müller (2009) |
| Eberswalde | 52.89 | 13.81 | Forest (coniferous) | 1978–1984 | 633 | 525 | 363 | Müller (2009) |
| Eberswalde | 52.89 | 13.81 | Forest (coniferous) | 1985–1989 | 625 | 525 | 476 | Müller (2009) |
| Eberswalde | 52.89 | 13.81 | Forest (coniferous) | 1990–1998 | 633 | 525 | 584 | Müller (2009) |
| Eberswalde | 52.89 | 13.81 | Forest (coniferous) | 1978–1984 | 633 | 525 | 443 | Müller (2009) |
| Eberswalde | 52.89 | 13.81 | Forest (coniferous) | 1985–1989 | 625 | 525 | 537 | Müller (2009) |
| Eberswalde | 52.89 | 13.81 | Forest (coniferous) | 1990–1998 | 633 | 525 | 625 | Müller (2009) |
| Castricum | 52.55 | 4.64 | Bare soil | 1941–1952 | 825 | 499 | 201 | Tollenaar and Ryckborst (1975) |
| Castricum | 52.55 | 4.64 | Bare soil | 1957–1966 | 893 | 503 | 205 | Tollenaar and Ryckborst (1975) |
| Castricum | 52.55 | 4.64 | Bare soil | 1972–1981 | 805 | 500 | 202 | Tollenaar and Ryckborst (1975) |
| Castricum | 52.55 | 4.64 | Bare soil | 1987–1996 | 887 | 503 | 192 | Tollenaar and Ryckborst (1975) |
| Castricum | 52.55 | 4.64 | Forest (coniferous) | 1941–1952 | 825 | 499 | 386 | Tollenaar and Ryckborst (1975) |
| Castricum | 52.55 | 4.64 | Forest (coniferous) | 1957–1966 | 893 | 503 | 680 | Tollenaar and Ryckborst (1975) |
| Castricum | 52.55 | 4.64 | Forest (coniferous) | 1972–1981 | 805 | 500 | 688 | Tollenaar and Ryckborst (1975) |
| Castricum | 52.55 | 4.64 | Forest (coniferous) | 1987–1996 | 887 | 503 | 764 | Tollenaar and Ryckborst (1975) |
| Castricum | 52.55 | 4.64 | Forest (deciduous) | 1941–1952 | 825 | 499 | 336 | Tollenaar and Ryckborst (1975) |
| Castricum | 52.55 | 4.64 | Forest (deciduous) | 1957–1966 | 893 | 503 | 519 | Tollenaar and Ryckborst (1975) |
| Castricum | 52.55 | 4.64 | Forest (deciduous) | 1972–1981 | 805 | 500 | 533 | Tollenaar and Ryckborst (1975) |
| Castricum | 52.55 | 4.64 | Forest (deciduous) | 1987–1996 | 887 | 503 | 534 | Tollenaar and Ryckborst (1975) |

[*] Derived from E-OBS.

[**] Mean of 24 lysimeters, minimum value 478 and maximum 614.

[***] Values digitized from Calder (1976)



**Table 2.** Data used in the Budyko-analysis (table continued because of double line-spacing). Units of fluxes are in mm/y.

| Site | Lat. | Lon. | Land use | Period | $P$ | PET[*] | ET | Reference/Source |
|------|------|------|----------|--------|-----|--------|-----|------------------|
| Plynlimon | 52.47 | −3.73 | Forest (coniferous) | 1974–1975 | 2300 | 552 | 999 | Calder (1976) |
| St. Arnold | 52.21 | 7.39 | Forest (coniferous) | 1969–1978 | 687 | 467 | 497 | Harsch et al. (2009) |
| St. Arnold | 52.21 | 7.39 | Forest (coniferous) | 1982–1991 | 765 | 490 | 582 | Harsch et al. (2009) |
| St. Arnold | 52.21 | 7.39 | Forest (coniferous) | 1995–2004 | 834 | 490 | 662 | Harsch et al. (2009) |
| St. Arnold | 52.21 | 7.39 | Forest (deciduous) | 1969–1978 | 687 | 467 | 364 | Harsch et al. (2009) |
| St. Arnold | 52.21 | 7.39 | Forest (deciduous) | 1982–1991 | 765 | 490 | 485 | Harsch et al. (2009) |
| St. Arnold | 52.21 | 7.39 | Forest (deciduous) | 1995–2004 | 834 | 490 | 638 | Harsch et al. (2009) |
| St. Arnold | 52.21 | 7.39 | Grassland | 1969–1978 | 687 | 467 | 343 | Harsch et al. (2009) |
| St. Arnold | 52.21 | 7.39 | Grassland | 1982–1991 | 765 | 490 | 332 | Harsch et al. (2009) |
| St. Arnold | 52.21 | 7.39 | Grassland | 1995–2004 | 834 | 490 | 427 | Harsch et al. (2009) |
| Rheindahlen | 51.14 | 6.37 | Grassland | 1983–1994 | 795 | 477 | 532 | Xu and Chen (2005) |
| Rietholzbach | 47.38 | 8.99 | Grassland | 1976–1985 | 1416 | 416 | 573 | Seneviratne et al. (2012) |
| Rietholzbach | 47.38 | 8.99 | Grassland | 1986–1995 | 1456 | 463 | 559 | Seneviratne et al. (2012) |
| Rietholzbach | 47.38 | 8.99 | Grassland | 1996–2005 | 1430 | 499 | 543 | Seneviratne et al. (2012) |
| Rietholzbach | 47.38 | 8.99 | Grassland | 2006–2015 | 1449 | 505 | 583 | Seneviratne et al. (2012) |
| Rotterdam | 51.93 | 4.47 | Urban | 2012 | 700 | 500 | 175 | Jacobs et al. (2015) |

[*] Derived from E-OBS.

[**] Mean of 24 lysimeters, minimum 478 and maximum 614.

[***] Values digitized from Calder (1976)



**Table 3.** Climate and land use contributions to changes in evapotranspiration and streamflow over the period 1960–2010. All units in $\text{km y}^{-1}$. For reference, $1 \text{ km y}^{-1}$ corresponds to an average discharge of $32 \text{ m}^3 \text{ s}^{-1}$. The total area with available data is $4{,}312{,}807 \text{ km}^2$.

| | Evapotranspiration | | | Streamflow | | |
|---|---|---|---|---|---|---|
| factor | positive | negative | net | positive | negative | net |
| Whole study domain | | | | | | |
| Land use | 53.6 | −9.26 | 44.3 | 9.3 | −53.6 | −44.3 |
| Precipitation | 89.2 | −48.2 | 41.0 | 165.3 | −76.0 | 89.3 |
| Potential evapotranspiration | 42.1 | −0.2 | 41.9 | 0.2 | −42.1 | −41.9 |
| Near-natural catchments (Stahl et al., 2010, and Fig. 6c,d) | | | | | | |
| Land use | 2.2 | −0.3 | 2.0 | 0.3 | −2.2 | −2.0 |
| Precipitation | 2.8 | −1.3 | 1.4 | 7.4 | −3.4 | 4.0 |
| Potential evapotranspiration | 2.6 | 0 | 2.6 | 0 | −2.6 | −2.6 |
| Rhine basin | | | | | | |
| Land use | 2.1 | −0.4 | 1.6 | 0.4 | −2.1 | −1.6 |
| Precipitation | 1.6 | −0.9 | 0.7 | 4.1 | −2.8 | 1.3 |
| Potential evapotranspiration | 2.7 | 0 | 2.7 | 0 | −2.7 | −2.7 |