# Peer review of "Climate change, re-/afforestation, and urbanisation impacts on evapotranspiration and streamflow in Europe"

_Hydrology and Earth System Sciences, 2018_

## Referee Comment (RC1) · Anonymous Referee #1 · 5 Feb 2019

The paper by Teuling et al., describes a simple modelling study where a one-parameter Budyko model is used to simulate changes in evapotranspiration and streamflow for most of Europe. It uses datasets on changes in precipitation, potential evapotranspiration (as function of air temperature), and landuse change as input and then simulates the change with the model. By reading the title, I Initially thought that this study is an attribution study which uses insitu data of evapotranspiration or runoff to attribute observed change to changes in climate or land use. But this is rather not the case, this study simply simulates changes and therefore the results should be discussed more cautiously.

Since the forest cover effect is hardcoded in the Budyko model, it will simulate changes in ET. However, this remains an extrapolation, which needs a better validation than what is now presented. The authors mention that the modelled ET average agrees well with the patters of GLEAM. Here I would like to ask the the authors to report statistics of this comparison. Then they also report a correlation with streamflow changes of r = 0.34, which corresponds to an explained variance of 12%, leaving 88% unexplained! Please show the scatterplot. Since there is a need for a validation of the model, I think that the model should be able to predict the observed streamflow changes better than a reference, for example using the changes in precipitation and maybe PET. Only if the Budyko model shows a higher skill I see justification to use that model and its change of the landuse parameterisation for the whole of Europe with confidence.

Land-use change is modelled by changing the parameter in the Budyko model using data from lysimeters. This is quite a central methodological step and ignores differences in scale of a lysimeter with that of a heterogeneous landscape. It also ignores that the parameter in the Budyko model can be different due to climatic variation, in particular the seasonality of rainfall to that of evaporative demand and the rainfall frequency. Jaramillo et al., 2018 HESS showed that there are increases in evaporative fraction, not explained by climate for many catchments in Sweden. Yet the link to changes in forest properties was rather weak. In contrast this study prescribes a distinct effect of forest age, hence there is a strong tendency that this study assesses the upper range of changes in water balance (if the HILDA database actually reflects the changes).

The choice of Thornthwaite method for PET is not acceptable for various reasons: a) It underestimates the evaporative demand (PET or Rn/L, see also van der Schrier 2011 or Maes et al., 2018). An annual average of PET of 700mm/yr for Southern Europe is far to low. That is why the authors need to scale it by an arbitrary factor aPET in the Budyko curve. b) Since it is a function of temperature only, it will be overly sensitive to warming trends which is arguably pretty strong for the considered period. It also

misses changes in shortwave solar radiation, see e.g. Wild et al., 2007. c) The authors argue for Thornthwaite because of data availability. However, there is data on sunshine duration / cloud cover. Furthermore, the diurnal temperature range correlates with solar radiation and has been used as a proxy for this, e.g. Wild et al., 2007, Makowski et al. 2008.

Apart from these major issues I enjoyed reading the paper. It is very well written, is well structured and has appealing figures. The topic is of high relevance for HESS. However, I believe that the validity of the Budyko approach needs to be demonstrated and therefore I recommend major revisions.

Minor Remarks:

Introduction, L20ff: it is argued that there are no sufficient studies which treat both landuse change and climate change on streamflow / ET. However, there are studies which indeed try to accomplish this, which I want to bring to the attention of the authors. For example Jaramillo et al., 2018 assessed changes in multiple catchments in Sweden. Renner et al., 2014 assessed observed changes of streamflow in East Germany. Lopez-Moreno et al., 2011 for catchments in Spain.

Figure 3: color of missing values (NA) should not be white, as indicated in the legend

Figures 6,7: there should be a color legend, a 3D color scheme on a map is a beautiful drawing but really difficult to grasp. What is the meaning of grey here? Similar magnitude of all drivers or a missing value? To what reference are the data scaled 2-98%, all of Europe?

The choice of rectangular sub-regions seems arbitrary to me. Why not use relevant river basins, where data is available to see if your prediction is indeed pointing in the right direction. For example on P9L10 it is mentioned that Scotland shows dramatic increases in streamflow, is this finding supported by observed changes?

Table 3: The units in the caption should be km^3/yr and not km/yr. In any case I would

[Figure]

prefer fluxes per unit area to allow comparison. Further I think that the total changes in Q / ET should be reported, not just the contributions.

References: Jaramillo, F., Cory, N., Arheimer, B., Laudon, H., van der Velde, Y., Hasper, T. B., et al. (2018). Dominant effect of increasing forest biomass on evapotranspiration: interpretations of movement in Budyko space. Hydrol. Earth Syst. Sci., 22(1), 567–580. https://doi.org/10.5194/hess-22-567-2018

Renner, M., Brust, K., Schwärzel, K., Volk, M., & Bernhofer, C. (2014). Separating the effects of changes in land cover and climate: a hydro-meteorological analysis of the past 60 yr in Saxony, Germany. Hydrol. Earth Syst. Sci., 18(1), 389–405. https://doi.org/10.5194/hess-18-389-2014

Makowski, K., Wild, M., & Ohmura, A. (2008). Diurnal temperature range over Europe between 1950 and 2005. Atmos. Chem. Phys., 8(21), 6483–6498. https://doi.org/10.5194/acp-8-6483-2008

Wild, M., Ohmura, A., & Makowski, K. (2007). Impact of global dimming and brightening on global warming. Geophysical Research Letters, 34(4), L04702. https://doi.org/10.1029/2006GL028031
* * *

---

## Short Comment (SC1) · 11 Feb 2019

We appreciate the detailed review by Anonymous Referee 1 and his/her generally positive impression of our work. Here we briefly respond to the more significant issues raised in the hope of some further discussion.

The reviewers' main comment is that the model requires a better validation. While our main approach was to present a framework that is constrained by observations in all aspects (in fact we are to first to combine available long-term observations made at lysimeter stations across Europe within a modelling framework – a fact that we will emphasize more in a revised version), we agree with the request for a more quantitative

validation against GLEAM ET and observed streamflow changes. We will look into this as soon as possible. But a perfect match or high explained variance should not be expected because a) our model has higher spatial resolution forcing input, b) our model simulates land use features such as cities that might not be well captured in other products (it should be noted that GLEAM is also a model product rather than an observation), and c) validation against a higher order statistic such as streamflow changes is more challenging (in fact very few studies do this) because changes might be induced by changes in measurement protocol, rating curve, weir conditions etc that are not captured by the model, so no high model performance should be expected (but we agree that it should be higher than observed P).

A second comment relates to the use of the Thornthwaite model for potential ET. We fully agree with the referee that this method should not be a default choice for modelling studies, and we have been explicit in pointing out the limitations of the method in the Methods and Discussion sections. The main reasons why we think the use is acceptable in this case are a) we prioritize maximization of the simulation period since the most significant land use changes occurred before the satellite era, and no gridded observational datasets are available that allow for any other than temperature-based PET methods to be applied before the 1950s, and b) the impact of changes in PET on streamflow and ET was found to be smaller than land use and precipitation changes, and since Thornthwaite likely over- rather than underestimates the true trend since the 1950s (i.e. before the strong dimming period), using another method will not affect these results. The referee indicates that "However, there is data on sunshine duration / cloud cover", but to our best knowledge such observational datasets are not available in gridded form from the 1950s onwards. However we do take the comments on the PET seriously and will thoroughly check our codes to ensure the values for southern Europe are at least correctly modelled.

The referee also stresses the impact of forest age on ET and movements in the Budyko space, citing the work of Jaramillo. We fully agree with this observation, which is also

why we decided to include the effect of stand age (see Figure 1). Maybe the referee missed this aspect of our model. Based on previous work on analysis of long-term data from forested lysimeters (Teuling et al., Vadose Zone J 2018, Fig. 5), it was concluded that the most significant changes in ET from forests occur between 0 and 20 years. Hence, we used 3 different parameters for each 10 years in line with the temporal resolution of the HILDA land use dataset (also 10 years). The work by Jaramillo, which we missed in the initial submission, will be referred to in a revised version as motivation for our approach to distinguish 3 different values for the Budyko parameter based on stand age.

---

## Referee Comment (RC2) · Anonymous Referee #1 · 19 Feb 2019

In order to follow the discussion, I want to briefly comment on the authors reply to my comments.

One of the main conclusions from the paper is that land-use change had a strong effect on streamflow and evapotranspiration at the European scale. This is quite a strong finding and it should be based on adequate model validation. Thereby I believe that the model should predict the streamflow changes better than a prediction simply based on precipitation and PET changes. I am positive that the authors agree with my argument and look forward to the results.

The authors argue that the Thornthwaite method is preferred because of data availabil-

ity of gridded temperature. I want to bring to attention that there is data on minimum and maximum temperature from EOBS which can be used to calculate PET with the Hargreaves formulation. An example for Europe is Spinoni et al., 2017 which use EOBS data and compare Hargreaves with Thornthwaite and modified Thornthwaite. They show that Thornthwaite underestimates PET compared to Hargreaves. As I argued in my first comment, this underestimation is problematic in the Budyko approach leading to exceedance of the energy limit. In addition to the EOBS data, CRU provides a 0.5° PET data product based on Penman-Monteith. See https://crudata.uea.ac.uk/cru/data/hrg/

Spinoni, J., Naumann, G., & Vogt, J. V. (2017). Pan-European seasonal trends and recent changes of drought frequency and severity. Global and Planetary Change, 148, 113–130. https://doi.org/10.1016/j.gloplacha.2016.11.013

---

## Short Comment (SC2) · 27 Feb 2019

Note to the editor and authors: As part of an introductory course to the Master programme Earth & Environment at Wageningen University, students get the assignment to review a scientific paper. Since several years, students have been reviewing papers that are in open online discussion for Copernicus Journals, and the top students in the class have been asked to submit their reports to the discussion in order to help the review process. While these reports are written in the form of official (invited) reviews, they were not requested for by the editor, and we leave it up to the editor and authors to use these reports to their advantage. We hope that these reports will posi-

tively contribute to the scientific discussion and to the quality of papers published. This report/review was supervised by Prof. Wouter Peters.

Introduction

The paper of Teuling et al. (2019) elaborates on the new gained insight of the impacts of decadal changes in climate and land use on the amount and distribution of water resources availability across Europe since the 1950s. Therefore, the Budyko model was combined with a high-resolution historical land use reconstruction and a gridded observation of key meteorological variables to simulate the distribution of green and blue water. Overall the simulations agree well with the observations. According to Teuling et al 2019, the results show strong shifts in continental-scale patterns of evapotranspiration and streamflow since the 1950s.

Overall merit

It is an engaging paper. Until now assessments of past and future changes in streamflow have either focused on land use, climate contributions or on smaller catchments under particular climate conditions. The combined method, used in this paper, checks the boxes of relevance and novelty and is therefore an addition to existing literature. The advantage of understanding past and future changes in water availability is that it can contribute to better water resources management and planning. The authors state that land use impacts and climate change impacts, should be accounted for in future to prevent regional over or underestimation of changes in water availability. The paper focuses on the understanding of the impacts of changes in climate and land use on the amount and distribution of water resources availability, which addresses to the scope of HESS (Hydrological and Earth System Sciences).

Overall, the paper is well written. The methods, results and discussion have a clear structure and are easy to follow. The different sections are described in a thorough manner to support their argumentation, without loosing themselves in the details. Although the paper is of quality, some choices in the methodology are in my opinion not

yet up to standard and are in need of adjusting or clarification. Therefore, some major and minor revisions are requested before acceptance. These are elaborated in the next section.

Critique

Major arguments

The first major comment is about the Thornthwaite method mentioned on page 5 line 1. It is used for calculating PET and requires only the temperature as input (Thornthwaite, 1948). Since there is a warming trend since the 1950s, this choice of method is questionable. Multiple studies, such as Trajkovic and Kolakovic (2009), have found that the radiation-based methods more accurate reproduce reference PET than temperature-based methods. Fisher et al (2011) mention that temperature based models estimated 20–30% less than the radiation based models averaged across all their researched sites. It was even stated: "The choice of evapotranspiration model and input data is likely to have a bearing on model fits and predictions when used in analyses of species richness and related phenomena at geographical scales of analysis" (Fisher et al., 2011). Shaw and Riha (2011) state that the Priestley–Taylor equation (a primarily radiation-based model) consistently explained more of the variation in PET than temperature-based methods. The paper of Teuling et al (2019) acknowledges that Thornthwaite method does not always gives the strongest increase in PET values in a warming climate. Considering that the paper aims to understand the effect of climate change on green and blue water fluxes, the effect of a warming trend on the calculated PET values should not be overlooked. The temperature-based PET values will affect the main part of the paper, since it in used in the Budyko model to determine how the average precipitation is portioned between evapotranspiration and streamflow. To improve the quality of the paper, please switch to a radiation based model or add substantial argumentation, on why they picked the Thornthwaite method to calculate the PET over radiation-based methods.

The second major comment is about the observations in this study, which come from lysimeter stations according to P5 Line 23. These lysimeters are assumed to behave similar to landscape elements of 10e6 m2. The locations of these stations, are not evenly spread throughout Europe but mainly constrained central-west, as can be seen in Fig A1. The model forcing is based on interpolated observation from weather stations. The paper states that local land cover impacts on climate, such as enhanced temperature or cloud formation, should not be represented in the forcing dataset. The stations should indeed be carefully selected. WMO (2003) states "Observations of evapotranspiration should be representative of the plant cover and moisture conditions of the general surroundings of the station". Still, the interpolation of lysimeter stations should be representative for the whole of Europe, can this be achieved if the stations are only concentrated in the central-west? It can result in incorrect values near the edges of the maps of Fig 2-7. Please expand the amount and the spread of lysimeter stations or otherwise show the statistics to support the used method.

The third major comment is about the temporal scale. In the method section on page 7 (line 7) it was stated that changes over the intermediate 10-year periods (1955–1965 and 2005–2015) were analysed. It was stated "the trends were generally found to be monotonic". Therefore they calculated 10-year climate averages. These were used to force the Budyko model and calculate changes in evapotranspiration and water yield, so it influences the main part of the paper. The simulated continental scale patterns depend on these 10-year climate averages. The choice of words on line 9: "the trends were generally found to be monotonic" raises questions. What were the exceptions? Did this choice of temporal scope have significant effect on the calculated changes in evapotranspiration and water yield? As Zang et al (2004) states, the climatic variables precipitation, temperature, solar radiation and humidity have a large spatial en temporal variability. They interact with the catchment characteristics such as vegetation cover, which is of interest for Teuling et al. Therefore, please choose a smaller temporal scale in which the trends are all found to be monotonic or show the statistics of the trends over the 10-year periods to verify the choice to average them.
The fourth major comment is about correlation mentioned on page 8 line 9. The paper mentions that their approach is able to reproduce the overall pattern of observed changes in streamflow. It was stated "In spite of the difference in units and the fact that individual basins might have shorter record lengths, the correlation in trends between the basins is 0.34." However, a correlation of 0.34 leaves room for questions, is this correlation sufficient? It means that a large part of the data remains unexplained. The paper states that, the validation shows that their simplified approach is able to capture continental-scale patterns in mean and changes of blue and green water fluxes. Can the correlation of the pattern of observed changes in streamflow be improved by adjusting the input, such as the PET values calculated with a radiation-based model (Considering my first comment)? Please change in input to optimize the correlation or show more elaborate statistics and argumentation on why this correlation is sufficient.

Minor Arguments

P1 Line 5: Please replace the term 'green and blue water fluxes' with evapotranspiration and streamflow, to make it understandable without having to read the introduction.

P8 Line 24: simulated ET is shown in figure 5b while it is referenced to 5a

P8 Line 25: Observed ET is shown in figure 5a while it is referenced to 5b

P9 Line 25: Table 3 list the Europe-wide changes not table 2

P12 Line 1: Change 'Therefor' into Therefore

P11 Line 3: 'WMO recommendations' please include a reference

P21 Fig 1: needs revising and clarification:

a. The caption does not fully describe what is displayed in the figure. Please elaborate on the w* values.

b. Yellow line is hardly noticeable, consider changing it to another colour to improve readability

c. The legend on the left indicates the colours orange, light and dark green. However, it does not include red and yellow, what do those colours indicate?

d. In the end of the results, it was mentioned that the colours indicate the forest stand age, this should also be mentioned in the caption

e. The caption should include describing the grey dashed line as energy limit, to improve the understandability.

P23 Fig 3: needs revising and clarification:

a. The missing values (NA) are indicated by the colour white, however white is already used to indicate another fraction. This brings confusion what the colour is indeed indicating. Please indicate the missing values with another colour.

P24 Fig 4: needs revising and clarification:

a. Fig 4b and 4d indicate the change for evapotranspiration and streamflow. The change is indicated with green and blue colours to match the evapotranspiration (green) and streamflow (blue). They mention in the caption that they chose to reverse the colour scheme on purpose. However, it works confusing and counterintuitive. My recommendation is to choose a different colour scheme's to match the change in both the figures, without green and blue, to avoid confusion.

P25 Fig 5: needs revising and clarification:

a. In P8L24 and P8L25 there are references to figure 5, I mention below that they reference to the wrong part of the figure. However, one can also consider keeping the reference in that way, and change the order in the figure. In 5a and 5c the observation-based ET are shown and in 5b and 5d the simulated ET is shown. When the simulated ET figures are switched to the left, it will fit more clearly in the story line.

References:

Fisher, J. B., Whittaker, R. J., & Malhi, Y. (2011). ET come home: potential evapotranspiration in geographical ecology. Global Ecology and Biogeography, 20(1), 1-18.

Prudhomme, C., & Williamson, J. (2013). Derivation of RCM-driven potential evapotranspiration for hydrological climate change impact analysis in Great Britain: a comparison of methods and associated uncertainty in future projections. Hydrology and Earth System Sciences, 17(4), 1365-1377.

Shaw, S. B., & Riha, S. J. (2011). Assessing temperature‐based PET equations under a changing climate in temperate, deciduous forests. Hydrological Processes, 25(9), 1466-1478.

Thornthwaite, C. W.: An approach toward a rational classification of climate, Geogr. Rev., 38, 55–94, 1948.

Trajkovic, S., & Kolakovic, S. (2009). Evaluation of reference evapotranspiration equations under humid conditions. Water Resources Management, 23(14), 3057.

World Meteorological Organization, 2003: Manual on the Global Observing System. Volume I, wMO-no. 544, geneva.

Zhang, L., Hickel, K., Dawes, W. R., Chiew, F. H. S., Western, A. W., and Briggs, P. R.: A Rational Function Approach for Estimating Mean Annual Evapotranspiration, Water Resources Research, 40, https://doi.org/10.1029/2003WR002710, 2004.

---

## Referee Comment (RC3) · Anonymous Referee #2 · 7 Mar 2019

This article presents and a framework to explain changes in evapotranspiration and streamflow since the 1950's in Europe. This is an ambitious attempt to provide answers to a question recurrently asked to hydrologists. The methodology is clearly related to the attribution problem, a topical current issue for which the scientific community is still far from being able to design a well-established methodology. This is why such studies require special attention and care with regards to hypotheses and uncertainties. To my opinion, the proposed study relies on too many hypotheses that are not tested/mentioned clearly and consequently, the results are uncertain and questionable. Two examples are discussed hereafter pointing out the (related) main problems of the paper: i) uncertainties are not stated and quantified and ii) the results are not

validated while they could partially be.

1. The revisited land-use dependent Budyko curves

The land-use attribution relies heavily on the Budyko curves depicted on Figure 1. First I did not understand why this Figure is not discussed in the results section. The authors did a great job in collecting these lysimeter data but the amount of data remains too limited to design the whole modeling framework. Some curves are adjusted on the basis of very few points, e.g. w* is calibrated on the basis of two points for urban areas and these two points are extracted from a unique site of Arnhem. How can we state that this parameter will be representative of all urban areas in Europe? Some land use classes present more experimental points but the w* fitting is far from being satisfying, with large uncertainties, no clear distinct w* values for some classes and again many points are related to the same environmental data (the 26 points originated from only four sites, Table 1). Given the multiple sources of uncertainties, the authors should consider to quantify the parametric uncertainty (the sensitivity of the results to w* fitted values) and should try to validate the fitted w* on independent data (e.g. the streamflow data, see next comment).

2. Validation of the attribution results

The authors propose a validation exercise using GLEAM product and streamflow from near-natural catchments. It should be stated that the comparison to GLEAM cannot be viewed as a strict validation since GLEAM relies on hydrological modelling (different to Budyko but still a model using P and PET inputs). The validation using streamflow time series is to my opinion the unique way to perform a real validation with independent data. To perform a rigorous validation studies, the authors could compute for each catchment the observed "Net" change and compare it with the Net change computed by the Budyko-curves. The authors have the material to perform such validation that will provide a clear diagnosis on the method used for attribution. At this stage, the attribution exercise is more a sensitivity analysis, which is not at the level of the ambitious

objects of the study.

3. Other comments

p.2 l.13-17: this is a repetition with previous sentences. p.3 l.28-29: I disagree with this statement. The impact of urbanization is probably the most sensitive land use change impact on hydrological processes and it is discussed in the early hydrological literature (Leopold, 1968). See also the large sample studies by DeWalle et al. (2000) and the recent reviews on this topic (Oudin et al., 2018; Salvadore et al., 2015). p.4 A discussion on the attribution problem is missing. There is a large existing literature on attribution studies in hydrology and I suggest that the proposed methodology be described upon the several existing attribution studies and associated methodology (see the reviews by Dey and Mishra, 2017 and Wang, 2014). p.6 l.1-5 Is it verified by local measurements of PET? I do not understand how the "c" linear factor might accounts for "all processes affecting yearly ET for tall vegetation". p.7 6-10: Using 10-yr periods to assess hydrological changes is too small with regard to natural climate variability. p.8 l.28-29: Please clarify the differences in units and how the correlation trends is calculated. Besides, I am not sure that correlation is the more adequate tool, maybe a contingency table would be more appropriate to compare the observed and simulated trends. p.10 l.24-25. Please modify the sentence and replace the term believe.

References

DeWalle, D.R., Swistock, B.R., Johnson, T.E., McGuire, K.J., 2000. Potential effects of climate change and urbanization on mean annual streamflow in the United States. Water Resour. Res. 36, 2655–2664. https://doi.org/10.1029/2000WR900134 Dey, P., Mishra, A., 2017. Separating the impacts of climate change and human activities on streamflow: A review of methodologies and critical assumptions. Journal of Hydrology 548, 278–290. https://doi.org/10.1016/j.jhydrol.2017.03.014 Leopold, L.B., 1968. Hydrology for urban land planning - A guidebook on the hydrologic effects of urban land

use (USGS Numbered Series No. 554), Circular. U.S. Geological Survey, Reston, VA. Oudin, L., Salavati, B., Furusho-Percot, C., Ribstein, P., Saadi, M., 2018. Hydrological impacts of urbanization at the catchment scale. Journal of Hydrology 559, 774–786. https://doi.org/10.1016/j.jhydrol.2018.02.064 Salvadore, E., Bronders, J., Batelaan, O., 2015. Hydrological modelling of urbanized catchments: A review and future directions. Journal of Hydrology 529, 62–81. https://doi.org/10.1016/j.jhydrol.2015.06.028 Wang, X., 2014. Advances in separating effects of climate variability and human activity on stream discharge: An overview. Advances in Water Resources 71, 209–218. https://doi.org/10.1016/j.advwatres.2014.06.007

---

## Referee Comment (RC4) · Anonymous Referee #3 · 8 Mar 2019

I see in the article some interesting aspects that contribute to the literature such as the constraining of the ET from the Budyko model by specific land use-dependent lysimetric data and a detailed analysis of land use changes across Europe, to calculate changes in ET and R. However, I found several weaknesses of the current version that need to be addressed.

1. You say you constrain the w parameter of Eq. 3 with observations of different land types. If I understand correctly, you constrain the w parameter in the locations where you have lysimetric measurements of ET and data on PET and P, and then apply that same w across all the spatial area of that specific land use/cover in combination with

local PET and P data to obtain local ET rates across that land use/cover extension. But the lysimetric observations as you mention, are located in a very small area of Europe. I think that extrapolating those w parameter values to regions like Northern Scandinavia and Iberian Peninsula and other Mediterranean areas is unrealistic. Can't you rely on the work by (Sterling et al., 2013) to improve that constraining exercise or other databases of ET rates? I also think that the land use categories used are to course and omit others such as open-water areas or reservoirs, etc.

2. I know that the authors are aware of that (Page 10, line 30), but I see that there is no differentiation between irrigated and non-irrigated agriculture. Studies have found continental (Wang and Hejazi, 2011) and worldwide (Jaramillo and Destouni, 2015) driving effects from irrigation on long term ET and ET/P, from a Budyko perspective, and that are evident even at the large-basin scale. I think that a differentiation between irrigated and non-irrigated crops is compulsory for the constraining of the w parameter and the estimation of ET for land use/covers. In the same way, I would say that some of the attribution to reforestation in Southern Europe can be actually irrigation or rain fed agricultural intensification. Please check.

3. Why are the authors using the blue/green water framework, if they are also combining the terminology with fluxes, etc. For instance, they use across the text the terms blue water, runoff, water yield (Page 2 line 6), which appear to be referring to the same. The manuscript needs to be consistent in this way and I would say that green and blue terminology is relevant only when water consumption is being assessed. If not in agreement, please justify the use of such terminology and also cite the main source for such (Falkenmark, 1997).

4. It appears that an impact on long-term water partitioning from less now cannot be neglected that easily as stated in Page 6. See (Berghuijs et al., 2014) that also uses a Budyko approach. So at least acknowledge that uncertainty.

5. The authors justify their work "In spite of the direct link between average green and

blue water fluxes, few studies have addressed changes in both fluxes simultaneously. However, they omitted many works precisely doing that: (Rost et al., 2008; Siebert and Döll, 2010). I also find missing important works on the effects of forest change across Europe and from a Budyko framework perspective that have been omitted here (Jaramillo et al., 2018; Renner et al., 2014). These four studies would for sure enrich the discussion in relation to the attribution of the observed R and ET changes to forest change in Europe. Their findings should also support several of the statements expressed by the authors and interpreted from their results.

References

Berghuijs, W. R., Woods, R. A. and Hrachowitz, M.: A precipitation shift from snow towards rain leads to a decrease in streamflow, Nat. Clim. Change, 4(7), 583–586, doi:10.1038/nclimate2246, 2014.

Falkenmark, M.: Meeting water requirements of an expanding world population, Philos. Trans. R. Soc. B Biol. Sci., 352(1356), 929–936, doi:10.1098/rstb.1997.0072, 1997.

Jaramillo, F. and Destouni, G.: Local flow regulation and irrigation raise global human water consumption and footprint, Science, 350(6265), 1248–1251, doi:10.1126/science.aad1010, 2015.

Jaramillo, F., Cory, N., Arheimer, B., Laudon, H., van der Velde, Y., Hasper, T. B., Teutschbein, C. and Uddling, J.: Dominant effect of increasing forest biomass on evapotranspiration: interpretations of movement in Budyko space, Hydrol Earth Syst Sci, 22(1), 567–580, doi:10.5194/hess-22-567-2018, 2018.

Renner, M., Brust, K., Schwärzel, K., Volk, M. and Bernhofer, C.: Separating the effects of changes in land cover and climate: a hydro-meteorological analysis of the past 60 yr in Saxony, Germany, Hydrol Earth Syst Sci, 18(1), 389–405, doi:10.5194/hess-18-389-2014, 2014.

Rost, S., Gerten, D., Bondeau, A., Lucht, W., Rohwer, J. and Schaphoff, S.: Agricultural

green and blue water consumption and its influence on the global water system, Water Resour. Res., 44(9), W09405, doi:10.1029/2007WR006331, 2008.

Siebert, S. and Döll, P.: Quantifying blue and green virtual water contents in global crop production as well as potential production losses without irrigation, J. Hydrol., 384(3–4), 198–217, doi:10.1016/j.jhydrol.2009.07.031, 2010.

Sterling, S. M., Ducharne, A. and Polcher, J.: The impact of global land-cover change on the terrestrial water cycle, Nat. Clim. Change, 3(4), 385–390, doi:10.1038/nclimate1690, 2013.

Wang, D. and Hejazi, M.: Quantifying the relative contribution of the climate and direct human impacts on mean annual streamflow in the contiguous United States, Water Resour. Res., 47(10), n/a–n/a, doi:10.1029/2010WR010283, 2011.

---

## Author Comment (AC1) · 8 Mar 2019

We thank the Anonymous Referee for the constructive comments and suggestions for additional references. We tried to be as complete as possible in covering the literature on several relevant aspects, but obviously we still missed some important papers. These will be included in a revised version. Concerning some of the points raised by the reviewer, I would like to briefly discuss the following:

- Values for parameter $w^*$. We will check the database of Sterling. However it should be stressed that all data have been collected based on the following cri-

teria: 1) fully homogeneous land cover, 2) multi-year records whenever possible (very few seem to exist for homogeneous urban areas, hence we used shorter records here which might be justified given the much shorter response times of the ET timeseries)., 3) robust measurement techniques (eddy covariance observations might lead to opposite land use effects of ET, see discussion in Teuling, VZJ, 2018). Unfortunately we were not able to locate any datasets satisfying all criteria in Northerly or Mediterranean parts of Europe. We are happy to receive any specific suggestions on datasets that do satisfy these criteria. Of course it should be noted that one of the key assumptions behind the Budyko framework is that the model parameter itself does not depend on climate (although it might depend on seasonality), and that extrapolations into either extremely water or energy limited regimes will not be very sensitive to exact values of the parameter.

- Land use classes. We choose to use a limited number of land use classes reflecting those that have the biggest known impact on ET (forest, cities). One of our reasons to exclude open water is that little is known about long-term controls of evaporation of water bodies. While it may seem attractive to assume that evaporation will take place at a potential rate above lakes, radiation or temperature are likely not the main drivers because of the high thermal inertia of water bodies and the important role of atmospheric stability. We preferred not to consider them rather then to make unconstrained and uncertain estimates. We will motivate this better in a revised version.

---

## Author Comment (AC2) · 8 Mar 2019

We thank the referee again for the useful suggestion. We will likely switch to the CRU PET product in a revised version, in spite of the disadvantage of a lower spatial resolution.

---

## Author Comment (AC3) · 8 Mar 2019

We thank Fleur Verwaal for her constructive comments. This is very much appreciated. As noted in the response to another reviewer, we will switch to a different PET formulation in a revised version. This should solve many of the issues raised. Concerning the lysimeter stations: unfortunately not many more stations exist, in particular not in North and South Europe. We use to Budyko model to interpolate and extrapolate using observed climate forcing. It should be noted that our lysimeter observations do span much of the European climate space, and that the exact value of the Budyko parameter impacts the results to a much smaller extent in strongly water-limited regions (since

all water will evaporate independent on land use). Concerning the temporal scale, we feel that our use of the word "generally" has been misleading. In fact we did not find any behavior in the intermediate periods that differed from the general trend/difference over the longer period. We will rephrase this sentence in a revised version to avoid confusion.

---

## Author Comment (AC4) · 8 Mar 2019

We thank the reviewer for the useful and constrictive comments, which will help us to improve the manuscript further. Below are some quick replies to some of the issues raised.

1. We agree there is a considerable degree of uncertainty in the exact value of the Budyko parameters used. However we also believe that constraining our model on urban data from 2 sites (the reviewer erroneously noted that we used a single site only, but Table 1 is spread over 2 pages and Rotterdam is in fact listed as

a second urban site) is better than not constraining it at all and assuming that model parameters are known which many other studies do. The low value of the Budyko parameter for urban land use is also in line with the common knowledge that urbanisation leads to a considerable increase in runoff. We will discuss our results considering the literature mentioned by the reviewer in a revised version.

2. The reviewer mentions that "the comparison to GLEAM cannot be viewed as a strict validation since GLEAM relies on hydrological modelling". We respectfully disagree. Not because we disagree with the observations that GLEAM is largely a model product, but with the fact that validation can also be performed on model simulations (of course this is generally less useful, but it classifies as validation nonetheless). We added the comparison because GLEAM is a de-facto standard of gridded ET estimates, so we believe it is relevant to confront our simulations with GLEAM. But we fully agree that this comparison should not be over-interpreted. In fact, we believe our model produces more realistic estimates over urban areas because of the higher resolution (1 km) and the fact that our Budyko parameter for urban areas has been constrained by observations of actual ET over cities. We will also revisit our validation with streamflow changes once we have redone our simulations with a different PET forcing (see replies to other reviews).

3. The reviewer also states that "the impact of urbanization is probably the most sensitive land use change impact on hydrological processes and it is discussed in the early hydrological literature (Leopold, 1968)". While we don't disagree with the importance of changes in urban area on streamflow (in fact this was one of our main motivations to carry out the work) and we value the contribution of Leopold to this area, we believe the size of the impacts are comparable to impacts of deforestation, which have been subject of extensive study since the beginning of the 20th century (see for instance the early deforestation experiment conducted between 1910 and 1926 at Wagon Wheel Gap, Colorado). The suggestion to

include a review on the history of attribution in hydrology is useful. We will make sure that the suggested references are included in a revision. Also we will consider the good suggestion of using a contingency table rather than correlation. In fact we had our reservations on whether the correlation was the best measure for the validation given the difference in units (note that the streamflow data used in Stahl et al. is not freely available, so we have to rely on the values of the changes per basin as used in the paper).

---

## Referee Comment (RC5) · Anonymous Referee #4 · 13 Mar 2019

General comments

This paper assesses the contributions of land use and climate changes to historical changes in streamflow and evapotranspiration in Europe. This is done using a stationary Budyko approach for water partitioning constrained by lysimeter observations and adopting historical land use reconstructions and gridded climate data at a high resolution of 1 km x 1 km. The resulting simulated changes in streamflow and evapotranspiration are in line with observed counterparts, although the comparison of streamflow changes is less straightforward. The contributions of land use change and climate change (through precipitation and evapotranspiration) are assessed for Europe and

analysed in detail for eight selected regions.

Overall, the paper is well written and presents interesting results for the European continent. The authors use informative and well-prepared figures to illustrate their results. Several issues need attention such as the use of the terms green and blue water, the use of one (high) value to adapt the potential evapotranspiration and the aggregation of positive and negative contributions of land use or climate change at continental or large river basin scale. These and other specific comments can be found below. The paper includes many typos, several examples and other technical corrections can be found below as well.

Specific comments

1. P2, L3-17: The terms green and blue water are not appropriately used here. The total evaporative flux also includes blue water from irrigation with surface water or groundwater (see e.g. Falkenmark, 2000; Oki and Kanae (2006); Falkenmark and Rockström, 2010) and hence the total evapotranspiration cannot be equated with the green water flux. Furthermore, the blue water flux does not only include streamflow (lines 15-16) but also groundwater flow (as briefly mentioned in line 5). The authors are suggested to remove the terms green water and blue water from the manuscript to avoid any confusion and to be consistent in terminology throughout the manuscript. There is no need to use the terms green and blue water (and water yield), since the focus is on streamflow and evapotranspiration.

2. P2, L11-12: Why is this in particular true for Europe? I would expect that uncertainties and limitations in observations and models in other parts of the world are at least comparable to those in Europe, but probably often larger.

3. P4, L30: The readability of section 2 and also section 3 can be improved by distributing the contents of these sections over a few sub-sections.

4. P5, L20-23: This sentence seems to be contradictory. The FLUXNET observations

are not used in this study, because they are assumed to be more reliable for long-term water balance assessments. However, this study also considers long-term water balances. This should be better explained.

5. P6, L10: A c-value of 1.8 is high and apparently seems to be used for all grid cells in Europe. This value implies that about 45% (0.8/1.8) of the energy used for evaporation is not included in the calculation of the potential evapotranspiration. Is this related to the simplified method (Thornthwaite) used to estimate the potential evapotranspiration? Which mechanisms (besides advection) are responsible for this? Is it reasonable to use the same c-value for all land use types? For instance, due to evaporation of intercepted water, you might expect higher c-values for forests compared to cropland and grassland. In summary, the use of a constant and high value for correcting the potential evapotranspiration seems to be doubtful and partly limits the conclusions which can be drawn based on this study.

6. P8, L5 and L14: The authors seem to mean something different with central-western Europe in these two lines, where firstly they seem to refer to Belgium and the Netherlands and secondly to Switzerland and Austria and parts of Germany and France. Try to be more specific here.

7. P8, L9-10: Is it logical that the mean evapotranspiration is highest due to pronounced topography? Although the term 'pronounced topography' is not completely clear, in general evaporation rates will decrease with altitude.

8. P9, 25-27: Is it sensible to determine the net effect of for instance precipitation by balancing positive contributions from the north with negative contributions from the south? The net effect obscures the real contributions and potentially associated problems; however, these net effects are an important element of the main conclusions and the abstract. I would recommend the authors to reformulate relevant parts of the manuscript and highlight positive and negative contributions rather than net contributions.

9. P10, L1-3: Is it useful to compare the sensitivity of streamflow to past land use changes with the effects of future changes? In order to interpret the differences between these two studies, the reader should at least have information on the magnitude of the future land use changes and the approach employed in the 2009 study. For instance, the way streamflow is determined in this study probably will be very different from the way streamflow has been determined in the 2009 study.

10. P11, L13-25: What is the role of other variables than temperature and radiation in the determination of PET (i.e. humidity and wind) and what is the effect of excluding these variables on the results?

11. P11, L26-28: The statement that socio-economic impacts relate more directly to blue water fluxes compared to green water fluxes is not correct. Green water is the main source of water to produce food, feed, bioenergy, etc. (e.g. Oki and Kanae, 2006) and as such changes in green water availability and fluxes will have a large socio-economic impact.

12. P25, Figure 5: Can streamflow be validated on the rate of change or only based on the patterns of change, since the units of c. and d. are different?

Technical corrections

1. P1, L12: 'Mediterranean' instead of 'Mediterrenean', see also e.g. page 2, line 22 and page 7, line 23.

2. P1, L15: The meaning of ET is not clear here.

3. P5, L12: "... the magnitude of this effect ..."; which effect is meant here?

4. P5, L17: 'coarse' instead of 'course'.

5. P6, L19: "... and lateral transport between ..."; between what?

6. P7, L10: 'purposes' instead of 'porpuses'.

7. P7, L23: 'Iberian' instead of 'Iberina'.

8. P7, L27: 'Romania' instead of 'Romenia'.

9. P8, L5: 'increase' instead of 'increased'.

10. P9, L1: 'separate' instead of 'separate', see also line 26 on this page.

11. P9, L1: The rescaling of the contributions is not clear to me.

12. P9, L25: 'Table 3' instead of 'Table 2'.

13. P9, L26: 'precipitation' instead of 'prepitation'.

14. P9, L34: '4 kmˆ3/year' instead of 4 kmˆ3'? And '-2 kmˆ3/year' instead of -2 kmˆ3'?

15. P10, L16: 'physical' instead of 'phyical'.

16. P26-27, Figure 6-7: 'Table 3' instead of 'Table 2'.

17. P28, Table 1: How is it possible to use a reference from 1975 to obtain data from periods until 1996? This needs to be adapted.

18. P28-29, Table 1-2: Which minimum and maximum values are meant for **? And what is the unit? Where can we find the *** in the tables?

19. P30, Table 3: 'kmˆ3 yˆ-1' instead of 'km yˆ-1'?

References

Falkenmark, M. (2000) Competing freshwater and ecological services in the river basin perspective. Water International, 25(2): 172-177.

Falkenmark, M. and Rockström, J. (2010) Building water resilience in the face of global change: from a blue-only to a green-blue water approach to land-water management. Journal of Water Resources Planning and Management, 136(6): 606-610.

Oki, T and Kanae, S (2006) Global hydrological cycles and world water resources.

Science, 313: 1068–1072.

---

## Author Comment (AC5) · 14 Mar 2019

The comment by the referee on the limited number of data points for urban areas (currently only for Rotterdam and Arnhem in The Netherlands) has motivated us to continue our literature search. We have found a study by Christen and Vogt (Int. J. Climatol. 24, 1395-1421) who report on long-term eddy covariance measurements over the Swiss city of Basel. They report an average ET of 300 mm/y. Combined with their value for precipitation (830 mm/y) and an estimated PET of 650 mm, this gives values for PET/P (0.84) and ET/P (0.36) that are in line with the Budyko curve fitted on the data points for Arnhem and Rotterdam. We will include this additional point in a

revised analysis. We thank the reviewer again for motivating us to provide as robust as possible estimates for the Budyko parameters in our model.

---

## Author Comment (AC6) · 14 Mar 2019

We thank the reviewer for providing a detailed and constructive review. We are happy to see that most comments are in line with comments provided by other reviewers. As one of the main comments, the reviewer questions the high value for w*. Given the PET estimates used, this value is needed to ensure all observations fall within the water- and energy limits of the Budyko space. In a revised version, we will likely switch to CRU PET, which will also slightly change the location and scaling in Fig. 1. However a scaling will still be needed since many recharge observations for forest are higher than local PET. The value should be interpreted as a conversion to the maximum ET that

would occur if the land use is (coniferous) forest. It has no impact on estimates for low vegetation since the estimate of actual ET is made on the combination of aPET and w*, which effectively cancels out the correction. We also agree with the suggestion to put more emphasis on the difference between positive and negative contribution rather than the net effect.

————————————————

---

## Author Comment (AC7) · 14 Mar 2019

We have looked into the use of CRU PET, and the values we find for the Iberian Peninsula (up to 1300 mm/y, see Fig. 1 which can be compared to our Fig. 2c) are closer to the values expected by the reviewer. This should solve some of the comments on our use of PET. However in spite of the higher PET values in Southern Europe, we will still need to apply a correction factor to local ET because some of the in situ measurements of ET at forested sites (Plynlimon, Castricum) have values that exceed the local PET. Allowing observations to fall outside the Budyko water and energy limits will lead to an underestimation of the true effect of changes in forest cover, which we want to avoid.

[Figure]

We are happy to hear if the reviewer agrees with the proposed use of CRU PET, which we can readily implement.

[Figure]

[Figure]

**Fig. 1.** Distribution of long-term average CRU PET for comparison with Fig. 2a.

---

## Author Response (AR1)

Dear Editor,

Hereby we submit the revised version of our manuscript entitled "Climate change, re-/afforestation, and urbanisation impacts on evapotranspiration and streamflow in Europe". We feel we have been served well in the review process with constructive and detailed comments provided by 4 anonymous reviewers, and an additional review from one of our own students from Wageningen. We take the large number of reviewers as a sign that the work has sparked interest. We are also pleased to see that no fundamental flaws were identified during the review process. All reviewers seem to agree that the manuscript is interesting, well written, and is in principle publishable. In response to the main criticism about the potential evapotranspiration used, we have gone through considerable efforts to redo all simulations with CRU PET. As a result, all figures have been updated and the numbers have changed slightly, but it should be noted the conclusions remain unchanged.

We hope that the rebuttal will be positively received by the reviewers.

Best regards, on behalf of the authors,

Ryan Teuling Bennekom, 7-6-2019

**Anonymous Referee #1**

Since the forest cover effect is hardcoded in the Budyko model, it will simulate changes in ET. However, this remains an extrapolation, which needs a better validation than what is now presented. The authors mention that the modelled ET average agrees well with the patters of GLEAM. Here I would like to ask the the authors to report statistics of this comparison. Then they also report a correlation with streamflow changes of r = 0.34, which corresponds to an explained variance of 12%, leaving 88% unexplained! Please show the scatterplot. Since there is a need for a validation of the model, I think that the model should be able to predict the observed streamflow changes better than a reference, for example using the changes in precipitation and maybe PET. Only if the Budyko model shows a higher skill I see justification to use that model and its change of the landuse parameterisation for the whole of Europe with confidence.

We agree with the importance of validation as stressed by the referee. In response, we have added a 4\*4 contingency table (which was suggested by Anonymous Referee #2 and has the function of a scatterplot) and a comparison of the trends in simulated Q, P and PET over the basins of Stahl et al. (2012). From this analysis (shown in a second new figure) we find that our simulation is closest to the observed change. Both P and PET have larger (wet) bias than the model. In order to compare trends over different units we considered the median change (over all basins) normalized by the IQR (Figure 7). We did not include a quantitative comparison with GLEAM because it is not our goal to match this product. In particular, for forest and urban areas (the focus of our study), we believe our model to be more accurate than GLEAM because it has been constrained by long-term water balance observations rather than being validated on eddy covariance data (as is the case with GLEAM). The comparison is for reference only. We have now made this more clear in the text: "Model simulations are validated and compared ..." and "It should be noted that this comparison is added for reference only and should not be seen as validation: GLEAM is not a strictly observational dataset, and it does not necessarily provide better long-term estimates of ET for forest and urban areas."

Land-use change is modelled by changing the parameter in the Budyko model using data from lysimeters. This is quite a central methodological step and ignores differences in scale of a lysimeter with that of a heterogeneous landscape. It also ignores that the parameter in the Budyko model can be different due to climatic variation, in particular the seasonality of rainfall to that of evaporative demand and the rainfall frequency. Jaramillo et al., 2018 HESS showed that there are increases in evaporative fraction, not explained by climate for many catchments in Sweden. Yet the link to changes in forest properties was rather weak. In contrast this study prescribes a distinct effect of forest age, hence there is a strong tendency that this study assesses the upper range of changes in water balance (if the HILDA database actually reflects the changes).

We disagree with the reference to Jaramillo et al. as used here. They find that: "... the positive residual effect occurred along with increasing standing forest biomass in the temperate and boreal basin groups, increasing forest cover in the temperate basin group and no apparent changes in forest species composition in any group". This is fully consistent with our approach of accounting for a) changes in forest cover at the 1 km2 resolution (much smaller than the basins considered in the Jaramillo study) and b) the use of different Budyko parameters for increasing stand age (biomass) as calibrated from unique long-term lysimeter experiments. Hence we believe our study is not at odds with Jamamillo et al. In the revised version, we now refer to this study and also mention seasonality and the synchronicity between precipitation and potential evapotranspiration explicitly as factors that might potentially affect w\* in the Discussion (where other factors were already mentioned). Indeed, there is a scale mismatch between the size of the lysimeters of 625 m2 do not represent the surrounding landscape (why would they have been constructed, operated and maintained for 50+ years if they would not be representative of the larger region?). We now mention the scale of the lysimeters more explicitly in the text ("area varies from 1 to 625~m2 for the larger lysimeters at Castricum")

The choice of Thornthwaite method for PET is not acceptable for various reasons: a) It underestimates the evaporative demand (PET or Rn/L, see also van der Schrier 2011 or Maes et al., 2018). An annual average of PET of 700mm/yr for Southern Europe is far to low. That is why the authors need to scale it by an arbitrary factor aPET in the Budyko curve. b) Since it is a function of temperature only, it will be overly sensitive to

warming trends which is arguably pretty strong for the considered period. It also misses changes in shortwave solar radiation, see e.g. Wild et al., 2007. c) The authors argue for Thornthwaite because of data availability. However, there is data on sunshine duration / cloud cover. Furthermore, the diurnal temperature range correlates with solar radiation and has been used as a proxy for this, e.g. Wild et al., 2007, Makowski et al. 2008.

We agree, and have redone the whole analysis using CRU PET as was suggested in the follow-up comment by this reviewer. This has however not significantly changed the results, nor the fact that scaling is needed to account for annual actual ET rates (including interception evaporation) that exceed PET.

Apart from these major issues I enjoyed reading the paper. It is very well written, is well structured and has appealing figures. The topic is of high relevance for HESS. However, I believe that the validity of the Budyko approach needs to be demonstrated and therefore I recommend major revisions.

Thank you for the kind words. With the simulations now based on CRU PET and additional validation we hope that we can convince the reviewer of the validity of our study.

**Minor Remarks:**

Introduction, L20ff: it is argued that there are no sufficient studies which treat both landuse change and climate change on streamflow / ET. However, there are studies which indeed try to accomplish this, which I want to bring to the attention of the authors. For example Jaramillo et al., 2018 assessed changes in multiple catchments in Sweden. Renner et al., 2014 assessed observed changes of streamflow in East Germany. Lopez-Moreno et al., 2011 for catchments in Spain.

We now cite the studies by Jaramillo and Renner. It was not clear which study by Lopez-Moreno et al. the reviewer was referring to as no reference was provided. However it should be noted that neither Jaramillo nor Renner discuss or even mention impacts on streamflow (only ET).

Figure 3: color of missing values (NA) should not be white, as indicated in the legend

**Fixed**

Figures 6,7: there should be a color legend, a 3D color scheme on a map is a beautiful drawing but really difficult to grasp. What is the meaning of grey here? Similar magnitude of all drivers or a missing value? To what reference are the data scaled 2-98%, all of Europe? The choice of rectangular sub-regions seems arbitrary to me. Why not use relevant river basins, where data is available to see if your prediction is indeed pointing in the right direction. For example on P9L10 it is mentioned that Scotland shows dramatic increases in streamflow, is this finding supported by observed changes?

A legend for a 3D colormap would also be 3D (a cube) so not easy to display. We believe this would only add to possible confusion rather than make the interpretation easier. Instead, we have opted for zooming in on (rectangular) regions which show strong changes of different composition, and show the individual contributions and corresponding colors so that the interpolation can be based on those. Since none of the other 4 reviewers had the same comment, we decided to keep the figure as it is (note that this figure was iterated and discussed extensively among the authors). Concerning the Scotland example mentioned by the referee, it should be noted that many of the patterns in the figure result from E-OBS rainfall trends which are known to be robust since they incorporate most available raingauge data. In the case of Scotland, the increasing precipitation is well known (see e.g. <a href="https://www.adaptationscotland.org.uk/why-adapt/climate-trends-and-projections">https://www.adaptationscotland.org.uk/why-adapt/climate-trends-and-projections</a>). Given that this region is generally energy-limited, it should not come as a surprise that the increasing P acts to increase Q rather than ET. Similar arguments hold for the other examples. To better explain the scaling we changed the sentence into "Each contribution is inversely scaled between the 2nd and 98th percentiles over Europe to reflect its relative importance".

Table 3: The units in the caption should be  $km^3/yr$  and not km/yr. In any case I would prefer fluxes per unit area to allow comparison. Further I think that the total changes in Q / ET should be reported, not just the contributions.

Caption is corrected. We added the total area of the region so that number can easily be converted to per unit area. We did not report the total changes in this table since our aim was to compare contributions.

**Fleur Verwaal**

The first major comment is about the Thornthwaite method mentioned on page 5 line 1. It is used for calculating PET and requires only the temperature as input (Thornthwaite, 1948). Since there is a warming trend since the 1950s, this choice of method is questionable. Multiple studies, such as Trajkovic and Kolakovic (2009), have found that the radiation-based methods more accurate reproduce reference PET than temperature based methods. Fisher et al (2011) mention that temperature based models estimated 20–30% less than the radiation based models averaged across all their researched sites. It was even stated: "The choice of evapotranspiration model and input data is likely to have a bearing on model fits and predictions when used in analyses of species richness and related phenomena at geographical scales of analysis" (Fisher et al., 2011). Shaw and Riha (2011) state that the Priestley-Taylor equation (a primarily radiation-based model) consistently explained more of the variation in PET than temperature-based methods. The paper of Teuling et al (2019) acknowledges that Thornthwaite method does not always gives the strongest increase in PET values in a warming climate. Considering that the paper aims to understand the effect of climate change on green and blue water fluxes, the effect of a warming trend on the calculated PET values should not be overlooked. The temperature-based PET values will affect the main part of the paper, since it in used in the Budyko model to determine how the average precipitation is portioned between evapotranspiration and streamflow. To improve the quality of the paper, please switch to a radiation based model or add substantial argumentation, on why they picked the Thornthwaite method to calculate the PET over radiation-based methods.

**We agree with this comment, which was also brought forward by other reviewers. We have now redone our analysis using CRU PET.**

The second major comment is about the observations in this study, which come from lysimeter stations according to P5 Line 23. These lysimeters are assumed to behave similar to landscape elements of 10e6 m2. The locations of these stations, are not evenly spread throughout Europe but mainly constrained central-west, as can be seen in Fig A1. The model forcing is based on interpolated observation from weather stations. The paper states that local land cover impacts on climate, such as enhanced temperature or cloud formation, should not be represented in the forcing dataset. The stations should indeed be carefully selected. WMO (2003) states "Observations of evapotranspiration should be representative of the plant cover and moisture conditions of the general surroundings of the station". Still, the interpolation of lysimeter stations should be representative for the whole of Europe, can this be achieved if the stations are only concentrated in the central-west? It can result in incorrect values near the edges of the maps of Fig 2-7. Please expand the amount and the spread of lysimeter stations or otherwise show the statistics to support the used method.

Apparently the reviewer got the wrong impression that we interpolate lysimeter data. This is incorrect. We optimize a model with lysimeter data that is subsequently for with gridded (interpolated) meteorological observations and used for the spatial prediction of water balance partitioning. We of course agree with the fact that the distribution of the stations is far from ideal, however this is clearly beyond our control. Long-term lysimeter data are rare (most stations are used for experiments rather than climate monitoring). We are among the first to synthesize data from most of the stations into a single modelling framework constrained as much as possible by high-quality observations of water balance partitioning. We believe this approach, which is different to most other studies, adds value to the existing literature on trends that are mostly model-based. It should also be noted that there is no evidence that our model would not work well in other regions, since we are in agreement with many other local studies (like Jaramillo et al. in Sweden, of course in more water limited regions our model would simply be constrained by the water limit which should not be a point of debate when irrigation is not considered). It is assumed that the Budyko framework (just like any other model calibrated on network data such as FLUXNET that does not have a uniform distribution over all climate zones) can be used to transfer local findings to other climate conditions (it should be noted that the Budyko framework in particular was designed to do just that). This is a central assumption in all Budyko studies, and it has long been known that this is a reasonable assumption.

The third major comment is about the temporal scale. In the method section on page 7 (line 7) it was stated that changes over the intermediate 10-year periods (1955–1965 and 2005–2015) were analysed. It was stated "the

trends were generally found to be monotonic". Therefore they calculated 10-year climate averages. These were used to force the Budyko model and calculate changes in evapotranspiration and water yield, so it influences the main part of the paper. The simulated continental scale patterns depend on these 10-year climate averages. The choice of words on line 9: "the trends were generally found to be monotonic" raises questions. What were the exceptions? Did this choice of temporal scope have significant effect on the calculated changes in evapotranspiration and water yield? As Zang et al (2004) states, the climatic variables precipitation, temperature, solar radiation and humidity have a large spatial en temporal variability. They interact with the catchment characteristics such as vegetation cover, which is of interest for Teuling et al. Therefore, please choose a smaller temporal scale in which the trends are all found to be monotonic or show the statistics of the trends over the 10-year periods to verify the choice to average them.

The 10 year period was chosen because land use changes in HILDA are reported at this resolution, and not finer. We did not find any major deviations in the general trend in the intermediate periods, so, therefore, we decided to focus only on the oldest and most recent periods covered by all datasets. However, in response to the reviewer (and to reviewer #2 who had a similar remark) we show the maps with 10-year advancing changes in P and PET in this rebuttal. We are happy to include a figure in the main manuscript if the reviewers consider it necessary.

The fourth major comment is about correlation mentioned on page 8 line 9. The paper mentions that their approach is able to reproduce the overall pattern of observed changes in streamflow. It was stated "In spite of the difference in units and the fact that individual basins might have shorter record lengths, the correlation in trends between the basins is 0.34." However, a correlation of 0.34 leaves room for questions, is this correlation sufficient? It means that a large part of the data remains unexplained. The paper states that, the validation shows that their simplified approach is able to capture continental-scale patterns in mean and changes of blue and green water fluxes. Can the correlation of the pattern of observed changes in streamflow be improved by adjusting the input, such as the PET values calculated with a radiation-based model (Considering my first comment)? Please change in input to optimize the correlation or show more elaborate statistics and argumentation on why this correlation is sufficient.

In response to this comment and similar comments by other reviewers, we have extended the validation and included 2 new figures.

**Minor Arguments**

P1 Line 5: Please replace the term 'green and blue water fluxes' with evapotranspiration and streamflow, to make it understandable without having to read the introduction.

Done

P8 Line 24: simulated ET is shown in figure 5b while it is referenced to 5a

**Corrected**

P8 Line 25: Observed ET is shown in figure 5a while it is referenced to 5b

**Corrected**

P9 Line 25: Table 3 list the Europe-wide changes not table 2

This should be Table 2, but since Table 1 needed to be split over 2 pages it received both nr 1 and 2.

P12 Line 1: Change 'Therefor' into Therefore

**Done**

P11 Line 3: 'WMO recommendations' please include a reference

We included a reference to WMO (1993)

P21 Fig 1: needs revising and clarification:

a. The caption does not fully describe what is displayed in the figure. Please elaborate on the w\* values.

Caption has been extended.

b. Yellow line is hardly noticeable, consider changing it to another colour to improve readability

We added the line for reference since the Castricum lysimeter station provides data for bare soil. However this is not used in the analysis since HILDA doesn't have a bare soil class.

c. The legend on the left indicates the colours orange, light and dark green. However, it does not include red and yellow, what do those colours indicate?

The figure mixes land use and stand age effects on w\*, hence we use different symbols and colors. To avoid confusion we have added colors for the urban (red) and bare soil (yellow) classes.

d. In the end of the results, it was mentioned that the colours indicate the forest stand age, this should also be mentioned in the caption

**Corrected**

e. The caption should include describing the grey dashed line as energy limit, to improve the understandability.

**Added.**

P23 Fig 3: needs revising and clarification: a. The missing values (NA) are indicated by the colour white, however white is already used to indicate another fraction. This brings confusion what the colour is indeed indicating. Please indicate the missing values with another colour.

**Figure has been revised.**

P24 Fig 4: needs revising and clarification: a. Fig 4b and 4d indicate the change for evapotranspiration and streamflow. The change is indicated with green and blue colours to match the evapotranspiration (green) and streamflow (blue). They mention in the caption that they chose to reverse the colour scheme on purpose. However, it works confusing and counterintuitive. My recommendation is to choose a different colour scheme's to match the change in both the figures, without green and blue, to avoid confusion.

We removed reference to blue and green water, yet since we believe it can help the interpretation if colours indicate change in the same direction we have kept the colorbars the same.

P25 Fig 5: needs revising and clarification: a. In P8L24 and P8L25 there are references to figure 5, I mention below that they reference to the wrong part of the figure. However, one can also consider keeping the reference in that way, and change the order in the figure. In 5a and 5c the observation based ET are shown and in 5b and 5d the simulated ET is shown. When the simulated ET figures are switched to the left, it will fit more clearly in the story line.

Good suggestion, we changed the panel from left to right and vice versa.

**P and PET changes in 10-year periods**

---

## Author Response (AR2)

One key limitation of the study is the validation of the model and its parameters derived from local lysimeter observations extrapolated to the whole of Europe. There is now a new figure which shows that trend direction and magnitude agree more than by chance. The sign of the trends, however, do not agree for all regions and it remains unclear for which reasons. I agree with the authors that a validation is difficult to achieve. I would, however, recommend to discuss these issues since this highlights that more research is needed to understand hydrological change.

We added the following sentence to the Discussion: "The observation that regional disagreement can exist between simulated and observed streamflow changes indicates that more research is needed to fully understand drivers of streamflow change at smaller (regional and catchment) scales."

Minor Issue: The Figs 9 and 10 show maps of the contribution to change. It is not clear which areas show missing values and which should show a change. For example, Norway is all gray, some parts of Sweden as well. The inset e) shows a violet color bit inside its all gray. Please clarify this.

We added the requested information to the caption of Figure 8 (numbered as 9 in the manuscript due to an error in the LaTeX package).